# Non-canonical interplay between glutamatergic NMDA and dopamine receptors shapes synaptogenesis

Nathan Bénac[1,7], G. Ezequiel Saraceno[1,7], Corey Butler [1,7], Nahoko Kuga[2,3], Yuya Nishimura[2], Taiki Yokoi [3], Ping Su[4], Takuya Sasaki [2,3], Mar Petit-Pedrol[1], Rémi Galland [1], Vincent Studer[1], Fang Liu[4], Yuji Ikegaya[2,5,6], Jean-Baptiste Sibarita [1] & Laurent Groc [1] ✉

Direct interactions between receptors at the neuronal surface have long been proposed to tune signaling cascades and neuronal communication in health and disease. Yet, the lack of direct investigation methods to measure, in live neurons, the interaction between different membrane receptors at the single molecule level has raised unanswered questions on the biophysical properties and biological roles of such receptor interactome. Using a multidimensional spectral single molecule-localization microscopy (MS-SMLM) approach, we monitored the interaction between two membrane receptors, i.e. glutamatergic NMDA (NMDAR) and G protein-coupled dopamine D1 (D1R) receptors. The transient interaction was randomly observed along the dendritic tree of hippocampal neurons. It was higher early in development, promoting the formation of NMDAR-D1R complexes in an mGluR5- and CK1-dependent manner, favoring NMDAR clusters and synaptogenesis in a dopamine receptor signaling-independent manner. Preventing the interaction in the neonate, and not adult, brain alters in vivo spontaneous neuronal network activity pattern in male mice. Thus, a weak and transient interaction between NMDAR and D1R plays a structural and functional role in the developing brain.

Understanding how developing neurons form functional networks underlying brain functions remains a central question in neuroscience. The vast majority of excitatory glutamatergic synapses are formed early in development during the synaptogenesis period. The glutamatergic NMDA receptor (NMDAR) strongly contributes to the early and late phase of synaptogenesis[1,2]. NMDARs are ionotropic glutamate receptor composed of two dimers of subunits, i.e. obligatory GluN1 subunits associated with GluN2 or 3 subunits, that are activated by agonist (glutamate) and co-agonists (glycine or D-serine)[2]. At developing synapses, NMDARs are among the first glutamatergic receptors to be detected, forming the so-called "silent" synapses that contain no/few labile AMPA receptors[3]. The NMDAR clustering constitutes thus an essential nucleation step for the early formation of synaptic sites[4]. Upon their activation at these immature unstable synaptic sites, NMDARs would flux calcium, activate signaling cascades, and stabilize AMPARs and scaffolding proteins. Other signaling molecules, such as

[1]Univ. Bordeaux, CNRS, IINS, UMR 5297, F-33000 Bordeaux, France. [2]Laboratory of Chemical Pharmacology, Graduate School of Pharmaceutical Sciences, The University of Tokyo, 7-3-1 Hongo Bunkyo-ku, Tokyo 113-0033, Japan. [3]Department of Pharmacology, Graduate School of Pharmaceutical Sciences, Tohoku University, 6-3 Aramaki-aoba, Sendai, Miyagi 980-8578, Japan. [4]Campbell Family Mental Health Research Institute, Centre for Addiction and Mental Health, University of Toronto, Toronto, Canada. [5]Center for Information and Neural Networks, Suita City, Osaka 565-0871, Japan. [6]Institute for AI and Beyond, The University of Tokyo, Tokyo 113-0033, Japan. [7]These authors contributed equally: Nathan Bénac, G. Ezequiel Saraceno, Corey Butler. ✉e-mail: laurent.groc@u-bordeaux.fr

adhesion receptors and gliomediators, have also been identified and investigated for their role in early synaptic assembly, maturation and maintenance[5–9]. Yet, the mechanism underpinning the very early clustering of essential synaptic membrane proteins, such as the NMDAR, remain however rather enigmatic.

The proteins present at the neuronal surface, i.e. the surfaceome, change over the course of brain development, with a high diversity at early stages[10]. Neurotransmitter receptors expressed at the plasma membrane of immature neurons are highly diffusive and poorly confined when compared to mature neurons[3,11,12]. Their cycling between intracellular and membrane pools is also upregulated at early stages[13]. Clustering the highly-diffusive NMDARs at early synaptic contacts would thus require some active and potent processes. Since early synaptic contacts are not yet equipped with intracellular postsynaptic scaffold apparatus[14,15], additional mechanisms are likely to contribute to the NMDAR early clustering. Besides their stabilization by intracellular proteins, NMDARs can directly interact with other surface proteins, including neurotransmitter and neuromodulatory receptors[16]. These surface protein-protein interactions stabilize and cluster NMDARs[16], promoting the possibility that such interactions play a role in synaptogenesis. Some of these interactors have been related to synaptogenesis and synaptic maturation processes, such as dopamine, Ephrin, and neuroligin receptors[17–19]. The dopamine receptor type I (D1R) and NMDAR interact through amino acid sequences located in their respective intracellular C-tails[20,21], and such an interaction strongly control the surface dynamics and distribution of both receptors[22–25]. However, even if the interaction between NMDAR and D1R have been extensively investigated for its functional role[20,23,26–30], precise biophysical characterization of the interaction in native condition is still lacking. Imaging techniques allowing both direct visualization and characterization of protein-protein interactions have been developed in heterologous cells[31–39] but no observation has yet been made in live neurons. Therefore, fundamental questions about their stability, their occurrence and regulation mechanisms must be answered to gain access to their biological roles. To address this key question, we here developed a multidimensional spectral single molecule localization microscopy approach (MS-SMLM) to directly visualize and biophysically characterize the interactions between NMDAR and D1R at the neuronal surface. We specifically investigated whether such putative interaction tunes NMDAR clustering during the period of synaptogenesis.

## Results

### Direct visualization and quantification of surface receptor-receptor interaction events in live neurons using MS-SMLM

Quantum dots (Qds)-based single nanoparticle experiments have been widely used to track the surface diffusion of receptors because of the pointing accuracy of the single molecule imaging and nanoparticle photostability[40]. We took advantage of these properties to concomitantly investigate the surface dynamics of dopamine receptor 1 (D1R) and GluN1 subunit-containing NMDAR (GluN1-NMDAR) after their labeling with Qds of different wavelengths onto cultured hippocampal neurons (Fig. 1a). We set a custom spectral microscope with a 4Pi configuration for versatile (2D + t + λ) MS-SMLM (Fig. 1b, c). Schematically, it is composed of two inverted microscope bodies precisely aligned one on top of the other: i) the bottom microscope performs state-of-the-art (2D + t) SMLM (here referred as "spatial") equipped with an azimuthal TIRF/HiLo illumination device, and ii) an upper microscope for spectral (λ) characterization using photons usually lost in traditional mono-objective configurations. The two microscopes were precisely aligned by translating the bottom microscope using a (x, y, θ, φ) stage placed below the bottom microscope. Such a geometry allows to perform 2D-localization using all photons collected by one high numerical aperture (NA) TIRF objective (×100 Oil, NA1.49) and determine the spectral signature of the detected

fluorophores using a second high-NA objective (×60, Water Dipping NA1) without compromising the localization performances. Two synchronized sensitive EMCCDs cameras allow the tracking of the bright Qds across the entire field of view of the EMCCDs (nearly 80 μm × 80 μm @ 30 Hz) using conventional filter sets and dichroic mirrors[41] (Fig. 1b). The precise localization (below the diffraction limit ~200 nm) of overlapping single emitters of each receptor (each receptor-Qd complexes were set at equivalent density) was determined using spectrally-informed multi-Gaussian fitting (Supplementary Fig. 1). This fitting allow us to estimate the lowest as possible distance between different emitters with a multi-fit error of 56 nm[41]. However, since we used a complex of antibody and Qd to track receptors, we arbitrarily set the search distance for putative interaction using the common cut-off of 100 nm (Fig. 1a)[34].

Transient events in which surface GluN1-NMDAR and D1R closely confine (i.e. below our search distance of 100 nm) were repetitively observed over time throughout the dendritic tree (Fig. 1d; Supplementary Fig. 1). These events were tracked over time and isolated (Fig. 1d, e). To define whether such an event was based on random colocalizations or specific interactions, we monitored the behavior of GluN1-NMDAR with either D1R wild-type (D1R-wt) or a truncated D1R in which the T2 sequence involved in the interaction was genetically deleted (Fig. 2a) as we expected both the occurrence and the duration of the events to be affected upon true interaction. Moreover, the reduced colocalization of D1R-dT2 and GluN1-NMDAR (defined as the fraction of GluN1 cluster area that overlap with D1R cluster) was also confirmed in hippocampal neurons through live surface immunostaining (Fig. 2b). The occurrence of interacting event in presence of D1R-dT2 was significantly reduced. In the control condition, 30% of the receptor localizations were interacting whereas it was decreased to 19% for GluN1-D1R-dT2. Furthermore, the average lifetime of the GluN1-NMDAR-D1R-dT2 interaction was significantly lower whereas the estimated dissociation rate or $K_{off}$ was significantly increased (Fig. 2c–f). However, the observed lifetime of the non-interacting tracks i.e. monomers remained unaltered in all conditions (Fig. 2d). The mean lifetime and estimated $K_{off}$ of GluN1 homodimers were also not altered by the presence of either D1R-wt or D1R-dT2 (Fig. 2d–f). These data indicated that our MS-SMLM approach is able to probe interactions between surface receptors in live neurons. The average lifetime of GluN1-NMDAR/D1R was $130 \pm 0.01$ ms and the estimated $K_{off}$ was $13 \pm 0.7$ s$^{-1}$. Spatially, D1R-GluN1-NMDAR interactions were highly labile and seem to occur randomly onto the dendritic shaft with no evidence of dedicated interaction hot spots (Supplementary Fig. 1d), therefore highlighting the stochastic nature of the interaction.

Because the GluN2A subunit also interact with the D1R C-tail through the T3 sequence (GluN2A-NMDAR::D1R)[20], we tested the putative role of such sequence on the GluN1-NMDAR-D1R interaction. For this, we used Förster Resonance Energy Transfer (FRET) imaging on COS-7 cells that express either GluN1-GluN2A (T2 and T3 domains) or GluN1-GluN2B (T2 only) subunits. We observe no differences in the GluN1-NMDAR-D1R FRET signals between these conditions, suggesting that the T3 domain has a negligible role (Supplementary Fig. 2). Furthermore, the TAT-T2 competing peptide disrupted the interaction between GluN1-NMDAR and D1R[20,23] whereas the TAT-T3 competing peptide failed to do so when compared to the TAT-non sense sequence (TAT-NS) (Supplementary Fig. 2). Thus, it appears that the T2 sequence plays a major and dominant role in the NMDAR-D1R interaction.

### GluN1-D1R interaction is increased in immature neurons in a CK1- and mGluR5-dependent process

Synaptogenesis is an intense and rapid phase that starts during the second week in cultured hippocampal neurons and during the first postnatal weeks in rodents. We defined two time-windows based on the developmental stages, i.e. the number of synapses, of hippocampal neurons, referred as "immature" and "mature": the immature window

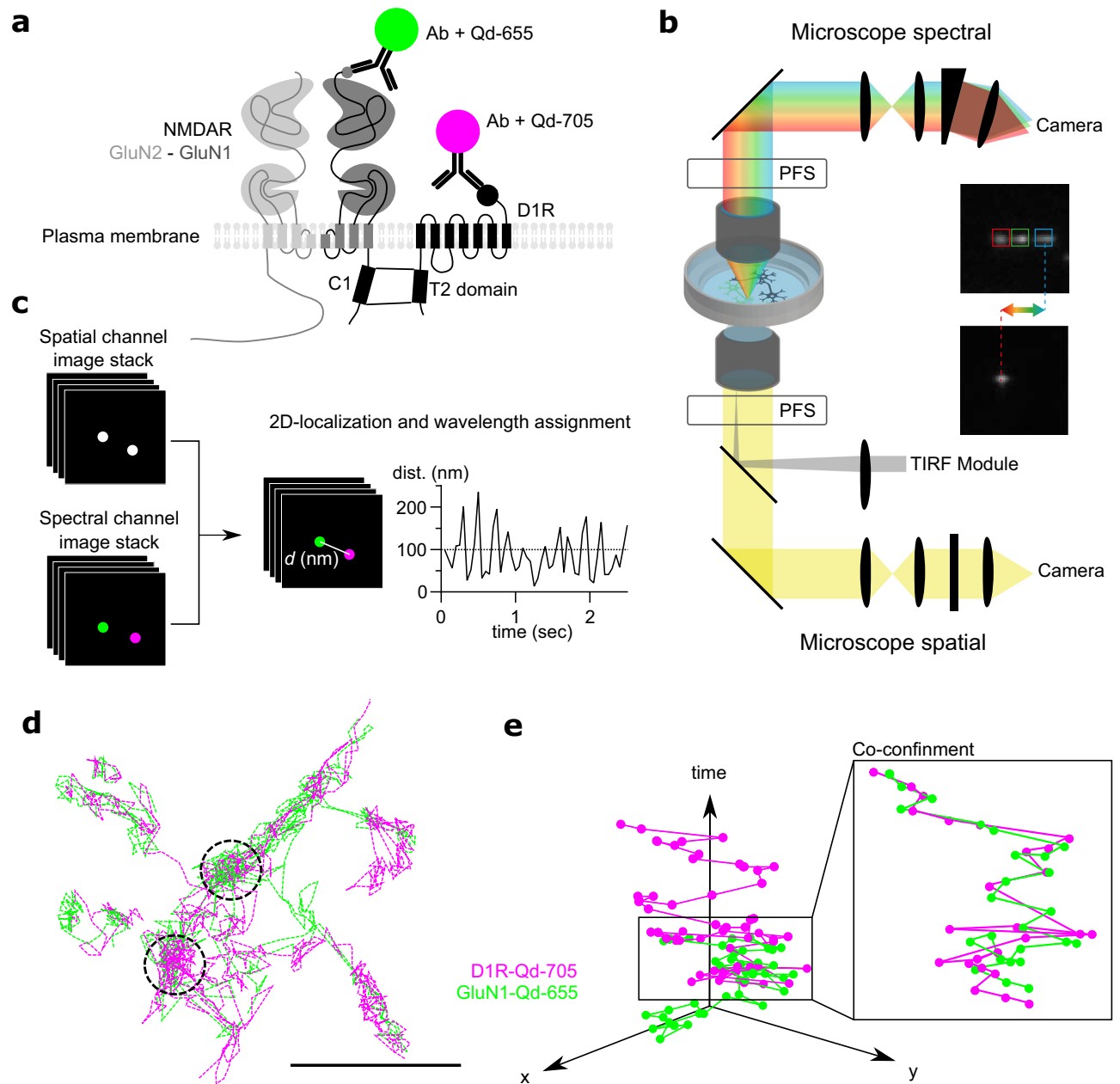

**Fig. 1 | Multidimensional spectral single molecule localization microscopy (MS-SMLM) principle. a** Experimental design of the single Qd tracking. Receptors were labeled with antibodies directed against extracellular tags. Qd-655 and Qd-705 were used to distinguish receptor types. The interaction between receptors occurs intracellularly at the T2 domain (C1 cassette of the GluN1 subunit). **b** Microscopy setting to perform MS-SMLM using two microscopes and cameras. PFS: perfect focus system. **c** SM-SMLM principle. **d** Representative reconstruction of GuN1-NMDAR and D1R surface diffusion, scale bar, 1 μm. **e** Example trajectories of one GluN1-NMDAR and one D1R laterally diffusing (x, y) onto the neuronal surface over time.

corresponds to days in vitro (DIV) 7 to 9 at the beginning of the synaptogenic period whereas the mature one (DIV 15 and above) corresponds to the end of the period (Fig. 3a, b). Both MS-SMLM imaging and immunocytochemical labeling demonstrated both qualitative and quantitative changes in the interaction properties between GluN1-NMDAR and D1R across development. The observed average lifetime of D1R-GluN1-NMDAR heteromers was significantly decreased at the mature stage compared to the immature one and the estimated $K_{off}$ were significantly higher at the mature stage when compared to the immature one. However, the observed average lifetime of the non-interacting tracks or monomers remained unchanged (Fig. 3d), highlighting the specificity of the interaction change. This observation at the single receptor level was confirmed by the immunolabelling of

surface D1R and GluN1-NMDAR since they highly colocalized at immature stage but rarely at mature one (Fig. 3f). We further ascertained this observation by measuring the in vivo level of D1R-NMDAR complex using co-immunoprecipitation assay with an efficient antibody directed against the GluN2A subunit (no current efficient antibody against GluN1 subunit for such Co-IP) (Fig. 3h). The level of co-immunoprecipitated D1R-GluN2A subunit complex in the rat hippocampus was twice higher at postnatal day (P) 8 when compared to P36 animals. This change was specific to the D1R-NMDAR complex as the previously defined D2R-GluN2B-NMDAR[42] complex level was unchanged across development (Fig. 3i). Collectively, these data indicate the surface interactions between GluN1-NMDAR and D1R are differentially regulated during neuronal development. This increased interplay was

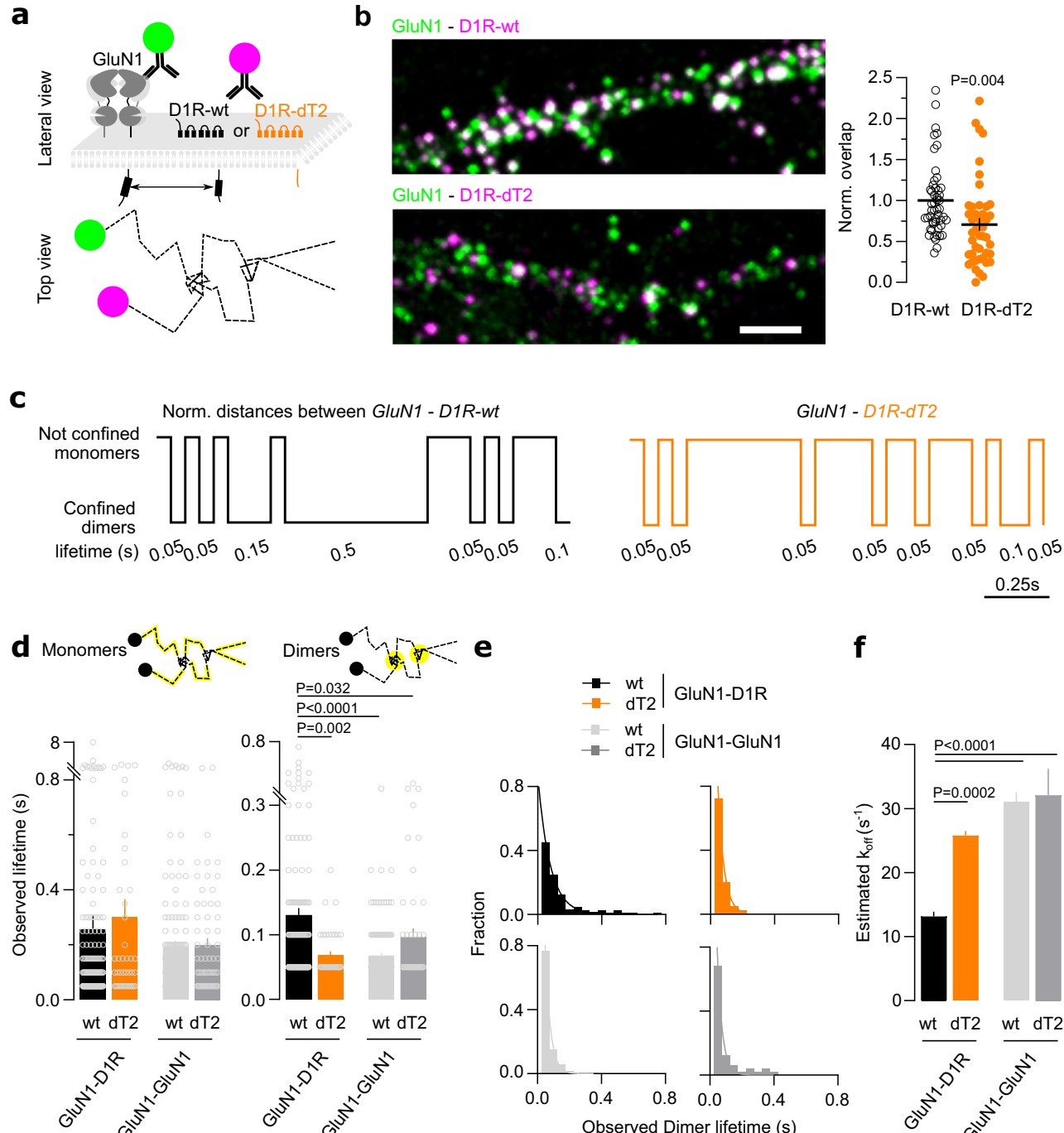

**Fig. 2 | MS-SMLM allows for the direct visualization and qualitative investigation of surface receptor-receptor interaction events in live neurons.**
**a** Experimental design of the surface single Qd tracking of NMDAR and D1R.
**b** Representative images with quantification of the normalized overlap between GluN1-NMDAR-D1R-wt (n = 50 neuronal fields) and GluN1-NMDAR-D1R-dT2 (n = 44 neuronal fields; two-tailed unpaired t-test). Data are presented as mean +/- SEM. Scale bar, 5 μm. **c** Representative normalized timeline of the distances separating one GluN1 from one D1R-wt or one GluN1 from one D1R-dT2. **d** Comparison of the average observed lifetime of the receptors (*left*) in non-confined space (monomeric state) for GluN1-D1R-wt (n = 168 events), GluN1-D1R-dT2 (n = 45), GluN1-GluN1 with D1R-wt (n = 165) or D1R-dT2 (n = 65); (*right*) in a co-confined space (dimeric state) between GluN1 and D1R-wt (n = 138), GluN1 and D1R-dT2 (n = 40), GluN1 and GluN1 expressed with D1R-wt (n = 159) or D1R-dT2 (n = 50; Kruskal-Wallis with Dunn's multiple comparisons test). Data are presented as mean +/- SEM. **e** Distribution and one exponential fit of the interaction events. **f** Comparison of the estimated $K_{off}$, i.e. dissociation rate (One-way ANOVA with Tukey's multiple comparisons test). Data are presented as mean +/- SEM. Source data are provided as a Source Data file.

also observed in cultured cortical neurons suggesting a shared mechanism for glutamatergic neurons (Supplementary Fig. 3).

We therefore investigated the mechanisms underpinning the surface interactions between GluN1-NMDAR and D1R in immature hippocampal neurons. As neuronal activity in the developing brain can tune synaptic maturation, neuronal activity was either up- or down-regulated by bath application of glutamate or tetrodotoxin (TTX), respectively. Silencing neuronal activity decreased GluN1-NMDAR-D1R co-localization whereas global activation of glutamate receptors increased it (Fig. 4a). Surprisingly, depolarizing neurons with KCl

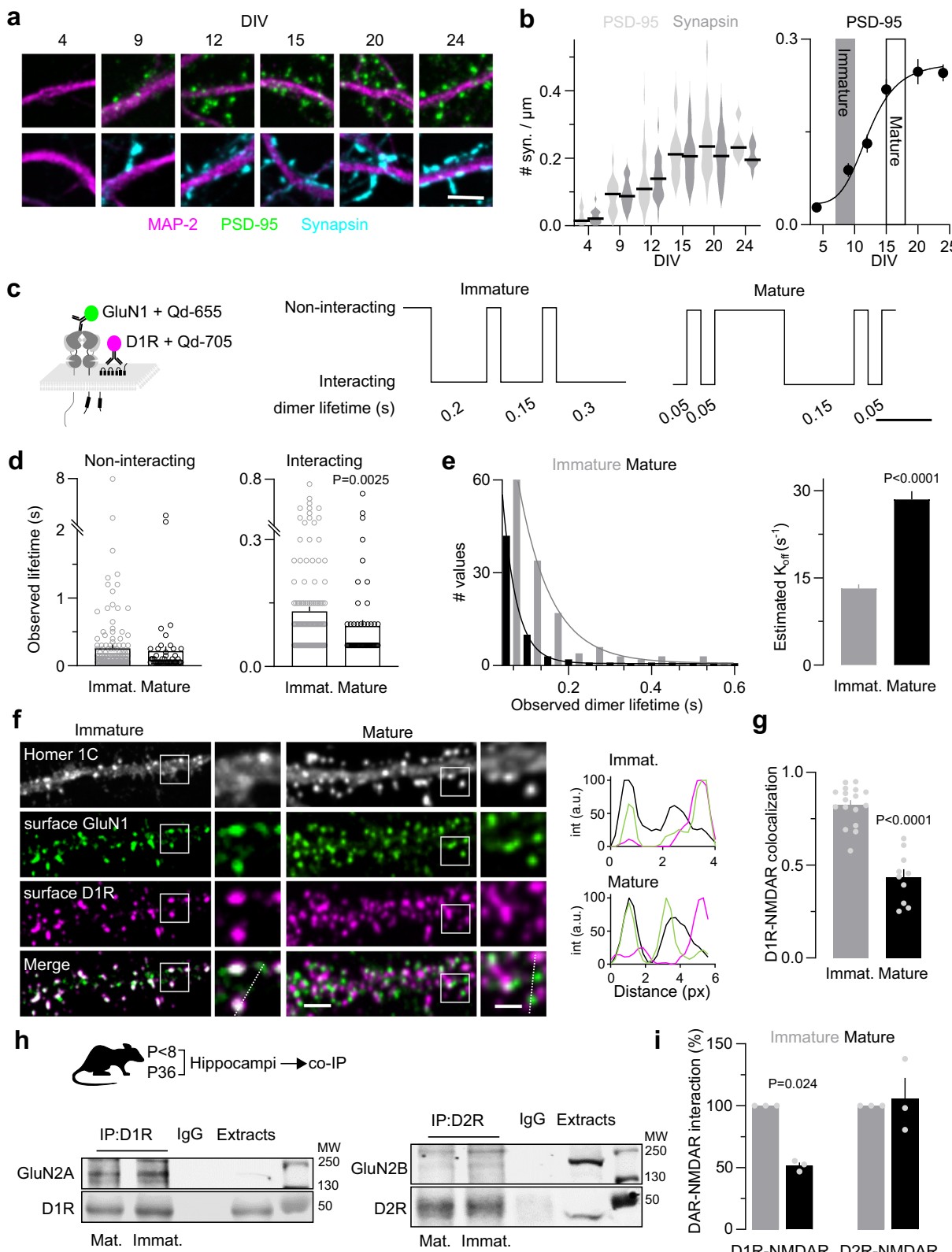

(50 mM), which increase neuronal firing, did not change GluN1-NMDAR-D1R co-localization (Fig. 4b). In fact, the glutamate-induced increase in GluN1-NMDAR-D1R colocalization was prevented by a mGluR inhibitor (LY341495) but not with antagonists of the AMPAR/kainate receptors (NBQX) or NMDAR (AP-5) (Fig. 4c), nor by co-applying NBQX and AP-5 together (Supplementary Fig. 4). The role of mGluR was evidenced by using the mGluR5 antagonist (MTEP), which

decreased GluN1-NMDAR-D1R colocalization (Supplementary Fig. 4). In addition, we could mimic the effect of ambient glutamate with the mGluR1/5 agonist DHPG (Fig. 4d). These data indicate that the GluN1-NMDAR-D1R interaction is regulated by neuronal activity and involve mGluR-dependent signaling processes.

To identify this pathway we focused on the interaction domain of the D1R with GluN1-NMDAR, the T2 domain[20]. The domain bears highly

**Fig. 3 | Increased dopamine-NMDA receptor interaction in immature neurons.**
**a** Representative image of hippocampal dendrites over in vitro development. Dendrites were labeled with MAP-2 (magenta), postsynaptic densities with PSD-95 (green), and presynaptic terminals with synapsin (blue). Scale bar, 5 μm. **b** (left) Quantification of the number of post-synapses (PSD-95) and pre-synapses (synapsin) at 4 DIV ($n = 18$ fields for both PSD-95 and synapsin), 9 DIV ($n = 17$ fields for both PSD-95 and synapsin), 12 DIV ($n = 30$ fields for PSD-95 and 29 for synapsin), 15 DIV ($n = 21$ for PSD-95 and 20 for synapsin), 21 DIV ($n = 21$ for PSD-95 and 20 for synapsin), 24 DIV ($n = 11$ for both PSD-95 and synapsin). Error bars represent the mean values; (right) non-linear fitting of the number of post-synapses over time, inflection point is at −12 DIV. Data are presented as mean +/- SEM. **c** (left) Experimental design. (right) Representative normalized timeline of the distance separating one GluN1 from one D1R in immature and mature neurons. **d** Comparison of the observed mean lifetime of (left) non-interacting GluN1-NMDAR-D1R in immature ($n = 168$ events) and mature ($n = 55$) neurons, (right) individual interacting events between GluN1 and D1R in immature ($n = 138$) and mature neurons ($n = 61$;

two-tailed Mann-Whitney U test). Data are presented as mean +/- SEM. **e** Distribution and one exponential fit of the interaction events between GluN1-D1R in immature and mature neurons with estimated $K_{off}$ (two-tailed unpaired t-test). Data are presented as mean +/- SEM. **f** Representative images of hippocampal dendrites on which surface GluN1-NMDAR (green), D1R (magenta), and Homer 1 C (white) were imaged in immature and mature neurons alongside corresponding intensity plots. Scale bar, 5 and 2 μm. **g** Quantification of the colocalization between D1R and GluN1-NMDAR in immature ($n = 17$ cells) and mature ($n = 10$ cells) neurons (two-tailed unpaired t-test). Data are presented as mean +/- SEM. **h** Experimental set-up and immunoblots. **i** Densitometric analysis of the levels of GluN2A and GluN2B co-immunoprecipitated by antibody directed towards D1R or D2R, respectively. The levels of D1R-NMDAR interaction were considered as the ratio of NMDAR co-IP with D1R-IP. Results are normalized to P8, 3 animals per condition (One-way ANOVA with Tukey's post hoc test). Data are presented as mean +/- SEM. Source data are provided as a Source Data file.

conserved sequences that are involved in the formation of non-covalent complexes through electrostatic interaction. Particularly, it contains a serine residue at the position 397 that can be phosphorylated by Casein Kinase 1 (CK1) which would lead to an increased interaction between D1R-GluN1-NMDAR[25,43] (Fig. 4e). We thus predicted that rising neuronal activity in immature neurons induces post-translational changes of the T2 domain in a CK1-dependent manner, modifying the interaction properties between D1R and GluN1-NMDAR. We first demonstrated the role for CK1 in regulating the interaction between D1R-GluN1-NMDAR in hippocampal glutamatergic neurons. To do so, we specifically inhibited CK1 as well as other protein kinases targeting other sites on the D1R and GluN1 C-termini, e.g. Casein Kinase 2 (CK2) and Calcium/Calmodulin-dependent Kinase II (CaMKII). CK1 inhibition disrupted GluN1-NMDAR-D1R co-localization whereas the inhibition of PKA, PKC, CK2, CaMKII or GRK2 had no effect (Fig. 4f). The role of CK1 was further confirmed by using another antagonist, i.e. IC-261, which also disrupted GluN1-NMDAR-D1R co-localization (Supplementary Fig. 4). To confirm that such unique post-translational modification of D1R could indeed strengthen the interaction between GluN1 and D1R, we generated a phosphomimic form of D1R, i.e. D1R-S397D. Both FRET on COS-7 cells and surface immunostaining on hippocampal neurons demonstrated an increased GluN1-NMDAR-D1R interaction with the phosphomimic form of D1R when compared to D1R-WT (Supplementary Fig. 4), a process that was activity-independent (Supplementary Figure 4). Together, these data support the view that D1R-GluN1-NMDAR clustering is tuned by changes in D1R-phosphorylation barcode in a CK1-dependent manner. An inhibition of CK1 (through CKI-7) together with the activation mGluRs (through DHPG) were sufficient to abolish the positive effect of DHPG on the interaction (Fig. 4g). Consistent with the prominent effect in immature neurons, the relative protein level of the subunit alpha of CK1, which is highly enriched within the hippocampus[44], was significantly increased at early stage, *i.e.* at the peak of D1R-GluN1-NMDAR interaction, in both hippocampal cultures and in the hippocampus in vivo (Supplementary Fig. 5). Collectively, these data show that D1R-GluN1-NMDAR interaction is favored during neuronal development by modification of D1R C-tail phosphorylation barcode by CK1 in a mGluR-dependent manner.

## GluN1-D1R interaction controls GluN1 surface organization

We next investigated the putative function of this increased interaction between GluN1-NMDAR and D1R onto immature neurons. Because it has been suggested that the interaction between GluN1 and D1R modulates their surface trafficking properties[23], we first tested whether the GluN1-NMDAR surface distribution is regulated by the receptor-receptor interaction. In these immature neurons, the prediction is that the receptor interaction favors the clustering of surface NMDARs, likely outside already formed early synaptic sites. First, we measured the areas of surface clusters of GluN1-NMDAR and tested whether

bigger NMDAR clusters were associated with a high content of D1R. In D1R-wt condition, GluN1-NMDAR cluster area positively correlates with the GluN1-NMDAR-D1R content (considered as the percentage of overlap between GluN1 and D1R) (Fig. 5a,b). Interestingly, the correlation was further increased in presence of phosphomimic D1R-S397D (Fig. 5b), whereas it was lost upon disruption of the interaction with D1R-dT2 (Fig. 5b). As D1R-dT2 is expressed for several days, we used a complementary approach to acutely disrupt GluN1-NMDAR-D1R interaction using a TAT-competing peptide that contains the T2 amino acid sequence (TAT-T2) or TAT-NS (control). Note that this peptide could partly interfere with the controversial D1R-D2R interaction that has been reported only in the striatum[45,46]. The competing peptide efficiently decreased the GluN1-NMDAR-D1R content (Supplementary Fig. 6). The inhibition of CK1 by CKI-7 produced a similar outcome (Supplementary Fig. 6). To better characterize this process, we used direct stochastic optical reconstruction microscopy (dSTORM) to probe the nanoscale organization of GluN1-NMDAR (Fig. 5c) as surface NMDARs are organized within nanodomains of 50-100 nm diameter[47]. To that extend, we co-expressed the GluN1 subunit with either D1R-wt, D1R-dT2 (no interaction) or D1R-S397D (strengthened interaction) (Fig. 5d, e). Although the number of nanodomains per clusters were not changed (Supplementary Fig. 6), the surface organization of GluN1-NMDARs between synaptic and extra-synaptic compartments were drastically impaired. GluN1-NMDARs are in the majority concentrated within the post-synaptic compartment with approximately one-third of the surface receptor pool being located extra-synaptically. As expected, in control condition surface GluN1-NMDARs were highly enriched at synapses (based on Homer 1 C) as demonstrated by a higher density of localization (number of localizations per nm²) in synaptic areas (Fig. 5d–f). This synaptic/extrasynaptic repartition was lost in presence of D1R-dT2 that equalized NMDAR nanodomain density (Fig. 5d–f). Remarkably, in presence of the phosphomimic D1R-S397D, the NMDAR distribution was shifted toward a higher extrasynaptic content (Fig. 5d–f). These data indicate that the interaction between NMDAR and D1R, which is located outside synapses (Fig. 5g)[23,48], regulate NMDAR nanoscale organization at extrasynaptic location (Fig. 5h).

We finally tested whether these clusters were functional and eventually recruited in early synaptic contact. For this, we co-expressed the calcium ($Ca^{2+}$) indicator GCaMP6f together with D1R-WT, D1R-dT2, or D1R-S397D and monitored the frequency of the NMDAR-mediated spontaneous $Ca^{2+}$ events in protrusions and onto the dendritic shaft of DIV 12 hippocampal neurons (Fig. 5i–k). At this stage, glutamatergic synapses are prominently located in protrusions (e.g. filopodia-like, spine; Supplementary Fig. 6). As expected, in the D1R-wt condition, the frequency of calcium transients was higher in protrusions than on dendrites (Fig. 5j, k), consistent with a higher amount of functional glutamatergic synapses in protrusions than

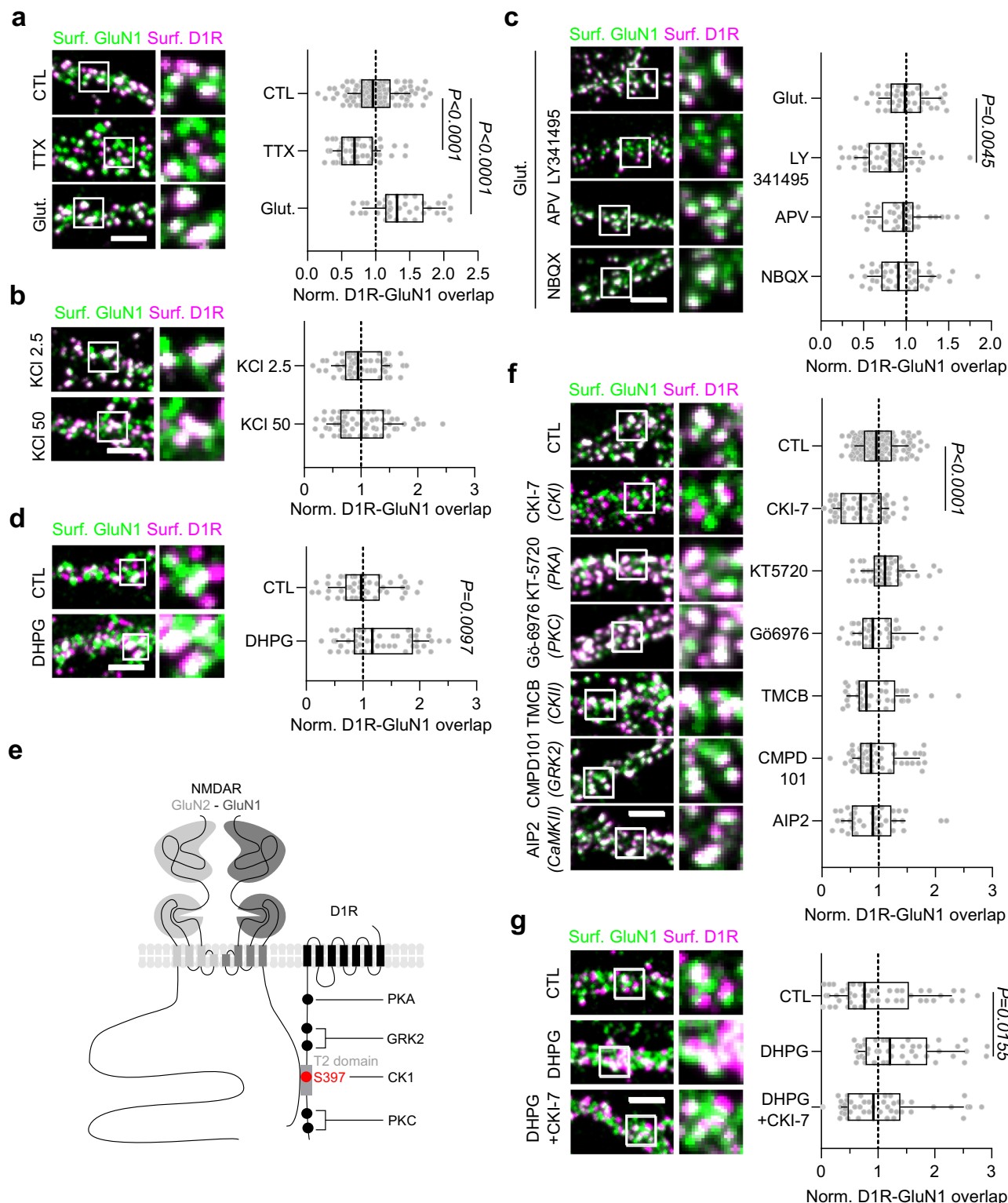

dendritic shaft. However, in the presence of the phosphomimic D1R-
S397D, the frequency of transients swapped and became higher on the
dendrites than in protrusions (Fig. 5j, k). In the presence of D1R-dT2,
the higher frequency of events in protrusions was not observed, with
equal event detection between protrusions and dendritic shafts
(Fig. 5j, k). Altogether, these data indicate that the interaction between
NMDAR and D1R favors the functional clustering of NMDARs in den-
dritic shaft of immature neurons, possibly reflecting a pro-
synaptogenic effect.

## Early in development, GluN1-D1R interaction tunes synaptogenesis

To directly address this possibility, we chronically disrupted D1R-
GluN1-NMDAR interaction by using TAT-based control (TAT-NS) or
competing (T2) peptides (see[23], Supplementary Fig. 2) during either an
early or late phase of synaptogenesis (Fig. 6a–d). Upon early chronic
disruption of the interaction with the TAT-T2 peptides, the number of
excitatory synapses (represented as the number of Homer 1 C cluster)
was significantly reduced by 15% (TAT-NS) whereas disruption of the

**Fig. 4 | GluN1-D1R interaction is activity-dependent and increased by the phosphorylation of D1R by casein kinase 1 (CK1). a–d** Representative images of hippocampal dendrites on which surface GluN1-NMDAR (green) and D1R (magenta) were labeled after exposures to various pharmacological treatments with respective quantification of D1R-GluN1-NMDAR mean overlap, (**a**) Buffer (CTL, $n = 105$ fields), TTX (1 μM, $n = 49$) and glutamate (50 μM, $n = 37$); (**b**) KCl 2.5 mM (CTL, $n = 55$) or KCl 50 mM ($n = 60$); (**d**) Buffer (CTL, $n = 47$) or DHPG (50 μM, $n = 51$); (**c**) glutamate alone ($n = 47$) or together with APV (50 μM, $n = 40$), LY-341495 (100 μM, $n = 52$) or NBQX (2 μM, $n = 43$) (**a**, **c** One-way ANOVA with Dunnett's post hoc test; b-d, two-tailed unpaired t-test). Scale bar, 5 μm. Results are normalized to CTL (**a**, **b**, **d**) or glutamate (**c**). **e** Cartoon illustrating putative D1R phosphorylation sites. **f** Representative images of hippocampal dendrites on which surface GluN1-NMDAR (green) and D1R (magenta) were labeled after exposures to various kinase inhibitors with respective quantification of the normalized GluN1-NMDAR-D1R mean overlap in control (CTL) condition ($n = 126$ fields) or following acute treatment with CKI-7 (100 μM, $n = 57$), KT-5720 (25 μM, $n = 49$), Gö-6976 (1 μM, $n = 41$), TMCB (5 μM, $n = 37$), CMPD101 (1 μM, $n = 50$) and AIP2 (1 μM, $n = 40$; One-way ANOVA with Dunnett's post-hoc test). Scale bar, 5 μm. **g** Representative images of hippocampal dendrites on which surface GluN1-NMDAR (green) and D1R (magenta) were labeled after treatment with buffer (CTL, $n = 48$ fields), DHPG alone (50 μM, $n = 44$) or together with CKI-7 (100 μM, $n = 45$) with respective quantification of D1R-GluN1-NMDAR mean overlap (One-way ANOVA with Dunnett's post-hoc test). Scale bar, 5 μm. Data are presented as box-and-whisker plots: line at median, IQR in box, whiskers represent 10–90 percentile. Source data are provided as a Source Data file.

interaction at the end of the synaptogenic period did not alter the number of excitatory synapses (Fig. 6b–e). Because synapses are more stable in mature neurons than in immature ones, we tested whether a longer exposition with the competing peptide modify the number of synapses in mature neurons. Yet, this number was not altered even after 8 days of treatment with control or competing peptide (Supplementary Fig. 7), therefore suggesting that the interaction is required for synaptogenesis only early in development. Consistently with an instrumental role, the strengthening of the interaction (obtained through the expression of the phosphomimic D1R-S397D) during that early stage significantly increased the number of glutamatergic synapses (Fig. 6g, h). Another hallmark of the maturation of glutamatergic synapses is the GluN2A/GluN2B subunit ratio that increases during development. We measured the synaptic content of GluN2A and GluN2B subunits in our experimental conditions. After an early disruption of the interaction, a premature high level of synaptic GluN2A-NMDAR and low level of GluN2B-NMDAR were observed, suggesting that the NMDAR-D1R interaction favors surface GluN2B-NMDAR functions (Fig. 6c–f). Altogether, these data indicate that D1R-GluN1-NMDAR interaction is necessary and sufficient at an early developmental stage to tune the formation and maturation of glutamatergic synapses.

An intriguing aspect of these observations comes from the fact that they were performed in hippocampal neuronal networks devoid of dopamine signaling. In this line, the level of dopamine in the developing hippocampus, i.e. the first two postnatal weeks, is extremely low. We confirmed this by immunolabelling tyrosine hydroxylase-positive (TH +) fibers in the CA1 area of P5 hippocampi. Fibers could be detected to a very low level when compared to other structures (e.g. striatum) and later stages (not shown) (Fig. 6i), indicating a low level of dopamine and noradrenaline. Since dopamine is present at low level in the neonate hippocampus, we investigated whether the synaptogenic role of the NMDAR-D1R interaction was preserved in developing hippocampal networks exposed to low level of dopaminergic fibers. To this end, we co-cultured hippocampal neurons with either hippocampal glutamatergic neurons (control, h-h) or dopaminergic-containing midbrain neurons (m-h) using microfluidic chip devices (Fig. 6j). We clarified the phenotype of cultured mesencephalic neurons originating from the ventral mesencephalon by immunolabeling with TH and dopamine hydroxylase (DBH), which convert dopamine to noradrenaline. We confirmed that the vast majority (90%) of our TH+ mesencephalic neurons are negative for DBH staining, thereby dopaminergic (Supplementary Fig. 8). Axons from TH+ dopaminergic neurons from chamber 1 were able to propagate into the hippocampal chamber (chamber 2) where they intermingled with glutamatergic neurons (Fig. 6j). Consistent the well-documented trophic effect of dopamine[17], hippocampal neurons co-cultured with dopaminergic neurons (m-h) exhibited higher dendritic arborization and complexity (Supplementary Fig. 8). After 12 days of hippocampal-mesencephalic co-culturing, the glutamatergic neurons that develop alongside dopaminergic fibers had higher synaptic density

compared to control (Fig. 6k, l), as expected from the well-established role of dopamine in network formation[17,49]. Yet, early disruption of the NMDAR-D1R interaction significantly decreased the number of excitatory synapses by 35%, a similar extent than in hippocampal neuronal network only (Fig. 6m, n). Collectively, these data demonstrate that in immature hippocampal networks the surface interaction between NMDAR and D1R tunes synaptogenesis in a dopamine signaling-independent manner.

## GluN1-D1R interaction is required for in vivo early hippocampal network activity

We next tested the role of this interaction in vivo in the developing hippocampus. Since the D1R-GluN1-NMDAR interaction is necessary and sufficient for the development of excitatory synapses, we predicted that the D1R-GluN1-NMDAR interaction contributes to the fine-tuning of early hippocampal network activity. First, the D1R-GluN1-NMDAR interaction was chronically disrupted in mice by intraperitoneal injections[50] of competing (TAT-T2) or control (TAT-NS) peptide at early stage (before postnatal day (P) 10) during the synaptogenesis window (Fig. 7). Hippocampal local field potential (LFP) activity was then recorded 1-2 days after the last peptide injection. Consistent with previous observations from the mouse cortex at postnatal transition[51], the hippocampus in non-anesthetized P7 head-fixed control mice spontaneously exhibited giant depolarizing potentials (GDPs), with an event frequency of $18 \pm 0.15$ per min. The chronic administration of competing TAT-T2 peptide, but not TAT-NS peptide, significantly reduced the GDP frequency (Fig. 7a, b). Note that there was no statistical difference between TAT-T2 and TAT-NS groups, due to the high variability of the TAT-NS mice. To complement this observation, we chronically disrupted the interaction during a similar postnatal period (P7-10) and recorded spontaneous hippocampal activity in urethane-anesthetized mice (P12). The control group spontaneously exhibited large-amplitude burst events (LB), with an event frequency of $1.8 \pm 0.26$ burst/min (Fig. 7c–e). In TAT-T2 mice, LB event frequency was significantly reduced and the inter-event intervals were significantly increased as shown by the rightward shift in the distribution (Fig. 7d). Thus, the D1R-GluN1-NMDAR interaction regulates some features of the spontaneous activity in the developing hippocampus. We then performed a similar series of experiments at the young adult stage (P30-35). At P35, LFP oscillatory patterns, and sharp wave and fast oscillatory ripples were detected in control freely-moving mice (Fig. 7f, g). The chronic administration of competing peptides had no effect on either LFP oscillatory pattern, or sharp wave and fast oscillatory ripple (Fig. 7f–i). Note that the hippocampal spontaneous activities at early (P7-12) and late (P35) stages were completely different and likely supported by different neuronal populations and processes. Yet, these results indicate that the D1R-GluN1-NMDAR interaction contributes to the fine-tuning of spontaneous hippocampal neuronal network activity early in development, with no detectable effect on the adult hippocampal spontaneous activity.

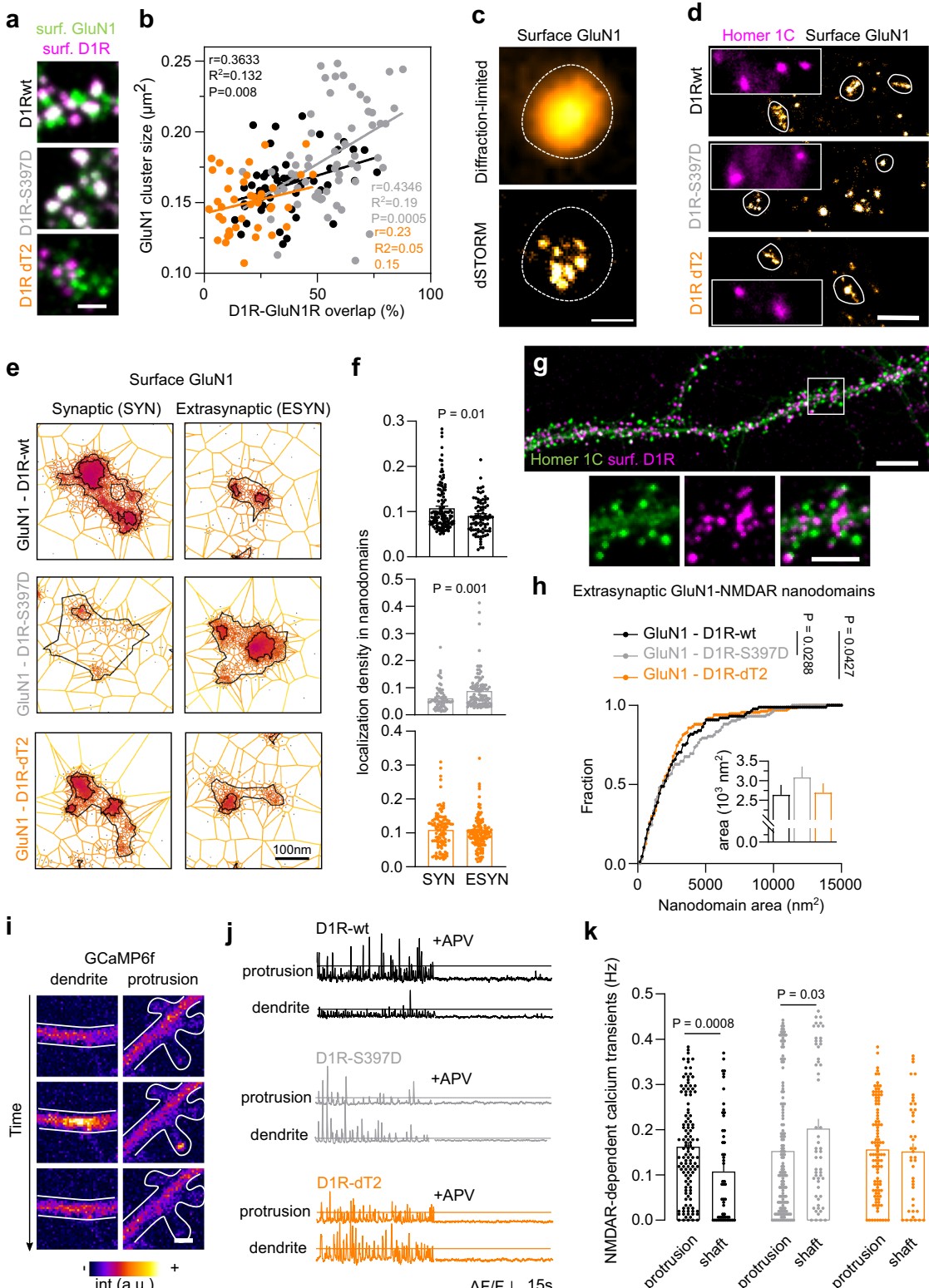

## Discussion

The development of imaging approaches such as single molecule-based microscopy, has unveiled with unprecedented spatio-temporal resolution, the dynamics and organization of receptors at the plasma membrane[52,53]. We provide the characterization of the membrane interaction between two prototypical and key receptors, i.e. the NMDAR and G-coupled dopamine D1R. At the surface of hippocampal neurons, their interaction lasts on average ~130 ms and occurs along

the dendritic tree, consistent with the stochastic surface dynamics of both receptors[12,23,54–56]. The interaction was qualitatively higher early in development, promoting the formation of NMDAR-D1R complexes in a mGluR5- and CK1-dependent manner. In immature hippocampal neurons, this interaction is stronger and favors membrane NMDAR clustering and synaptogenesis in a dopamine receptor signaling-independent manner. Thus, we unveil a non-canonical interplay between NMDAR and D1R, demonstrating that a weak and transient

**Fig. 5 | Surface interaction with D1R shapes GluN1 nano-organization and clustering. a** Representative image of hippocampal dendrites on which surface GluN1-NMDAR (green) and D1R (magenta) were labeled from D1R-wt, D1R-S397D (grey), or D1R-dT2 (orange) expressing neurons from 3 independent experiments. Scale bar, 2 μm. **b** Correlation between the size of GluN1-NMDAR cluster and the overlap between GluN1-NMDAR and D1R when co-expressed with D1R-wt, D1R-S397D or D1R-dT2. P-value were calculated using a two-sided *t*-test. **c** Example of diffraction-limited and super-resolution images of surface GluN1-NMDAR. Scale bar, 300 nm. **d** Representative images of super-resolved surface GluN1-NMDAR and diffraction-limited Homer 1 C staining. Scale bar, 1 μm. **c, d** Representative images from 4 independent experiments. **e** Representative clustering images obtained with SR-Tesseler software. Scale bar, 100 nm. **f** Comparison of the density of localizations per nanodomains inside and outside synapses when GluN1 is co-transfected with either D1R-WT (*n* = 7 cells), D1R-dT2 (*n* = 7) or D1R-S397D (*n* = 6;

two-tailed Mann-Whitney U test). Data are presented as mean +/- SEM. **g** Representative immunofluorescence image of surface D1R (magenta) and Homer 1 C (green) from 3 independent experiments. Scale bar, 10 and 5 μm. **h** Cumulative distribution of the area in nm² of extra-synaptic GluN1-NMDAR nanodomains synapses when GluN1 is co-transfected with either D1R-WT (*n* = 88 nanodomains), D1R-dT2 (*n* = 136 nanodomains) or D1R-S397D (*n* = 102 nanodomains; two-tailed Kolmogorov-Smirnov test). Bar graphs represent mean +/- SEM. **i** Representative GCaMP6f-fluorescence images from 3 independent experiments. Scale bar, 2 μm. **j** Representative NMDAR-mediated Ca²⁺ signals, scale is 0.05 (D1R-WT and -dT2) or 0.2 (D1R-397D) ΔF/F. **k** Comparison of the NMDAR-mediated Ca²⁺-transient frequency in protrusions and dendrite when GluN1 is expressed together with D1R-wt (*n* = 153 spines and 54 shaft), D1R-S397D (*n* = 145 spines, 56 shafts) or D1R-dT2 (*n* = 105 spines, 45 shafts; two-tailed Mann–Whitney U test). Data are presented as mean +/- SEM. Source data are provided as a Source Data file.

interaction between receptors can have a major structural and functional role for synaptogenesis in the developing hippocampus. As a consequence, the interaction modulates in vivo some features of the neonate spontaneous hippocampal network activity. The lack of effect on the adult spontaneous hippocampal activity does not preclude a functional role of the interaction in other network activities, specific tasks, and behaviors. Future studies will likely tackle these important questions.

Defining the properties of protein-protein interaction at the surface of living neurons has proven to be technically challenging. Commonly used methods to investigate receptor complexes in the brain, such as co-immunoprecipitation and proximity ligation assay (PLA), provide valuable information on the overall amount of hetero-receptor complexes present in a given structure at a given time but lack insights on the qualitative nature of the interaction between different receptors (for review, see[16]). For instance, critical parameters such as their occurrence or duration in the native environment have remained unknown. On the side note, the existence of membrane receptor heterocomplexes is still vividly debated (e.g. D1/D2 complex[45,46]). Thanks to the development of a custom MS-SMLM we were in the position to provide the first characterization of the interaction between surface GluN1-NMDAR and D1R at the single molecule level with nanoscale resolution in live neurons. The observed average lifetime of the GluN1-NMDAR-D1R interaction was $130 \pm 0.01$ ms, with a dissociation rate of $13 \pm 0.7\ \text{s}^{-1}$. These values are in a similar range to the ones defined from surface GPCR homomers in heterologous cells[31–33,35,36]. Finally, the D1R-NMDAR interactions were detected whole along the dendritic tree between receptors that stochastically cross each other since we didn't observe directed motion, or attraction, between receptors before/after the interaction. When single diffusing NMDAR and D1R get closer than 1 μm we observed that, on average, 30% of their localizations fall into the interaction area. At first glance, interactions do not appear to be clustered in specific areas, although we cannot rule out that specific membrane and/or intracellular subdomains favor the likelihood of interactions.

The strength of the interaction is finely regulated during development. Quite strikingly, we observed that during a rather limited developmental time window (DIV 10 to 15; doubling of glutamate synapse numbers) the GluN1-NMDAR-D1R interaction was twice stronger at the early time point, suggesting that important molecular cascade change during the synaptogenesis period. We identified on the D1R C-terminus a serine residue at position 397 that is regulated by CK1 and regulates the receptor interaction. Although weak and transient, the NMDAR-D1R interaction plays an instrumental role in clustering membrane NMDAR and promoting synaptogenesis in a mGluR-dependent process. Together with previous evidence showing that mGluRs are required for the experience- and activity-dependent changes in NMDAR transmission (i.e. GluN2A/GluN2B ratio) during development[57], our data fuel thus a developmental model in which ambient glutamate activates mGluRs that will locally favor NMDAR-

D1R interaction, NMDAR clustering, calcium influx, synaptogenesis, and synaptic maturation through regulation of the GluN2A/GluN2B ratio. In the absence of D1R (D1R knock-out mice), synaptogenesis and spinogenesis are expectedly strongly reduced[58]. The NMDAR-D1R interaction occurs early in development when the dopaminergic innervation of the hippocampus is rather scarce[59] supporting a preponderant role of protein-protein interaction independent of the presence of dopamine. The classical dopamine intracellular signaling cascade likely becomes prominent at adulthood when dopamine signaling controls hippocampal synaptic plasticity and cognitive functions[60–70]. The low level of dopamine in the neonate hippocampus further strengthens the NMDAR-D1R interaction because dopamine receptor activation drastically reduces this interaction[20,23]. In non-physiological conditions, an upregulation of dopamine levels early in development could thus strongly impact synaptogenesis and network formation. Consistently, in the brain of dopamine transporter knock-out mice the upregulated level of dopamine reduces the formation of synapses and spines[71]. The synaptic maturation of the GluN2A/GluN2B ratio is also corrupted in the brain of pups with elevated dopamine levels following mother exposition to cocaine, a deficit that could be rescued by positive modulation of mGluR[72]. Thus, low dopamine levels likely favor dopamine-NMDAR crosstalk and its function whereas high levels of dopamine activate dopamine receptors and classical GPRC signaling but shut-off the dopamine-NMDAR direct crosstalk.

Surface NMDAR interacts with other neuromodulatory GPCRs, such as cholinergic, adrenergic, or histaminergic ones[16] that may be involved in the newly-described protein-protein interplay. The size and composition extent of the described protein-protein complex is thus possibly larger. In addition to other monoamine receptors, NMDARs can interact with, for instance, ion channels (e.g. BK, TRPM[73–75]; and adhesion receptors, providing an additional layer of complexity on the composition of such a putative complex, while also highlighting its broad and strategic potential for regulating of G-protein signaling, protein kinase/phosphatase, agonist-induced ionotropic transmission, potassium currents and dendritic excitability. Furthermore, we demonstrate that the D1R-NMDAR direct crosstalk at early stage occurs in absence of dopamine, consistent with its low level in the hippocampus. Such non-canonical process echoes previous evidences demonstrating the non-canonical and functional interplay between the ghrelin receptor (GHSR1a) and D1R, which are both present in hippocampal neurons[76], but operate in a ghrelin-independent manner since ghrelin, the agonist of GHSR1a, is not present in the hippocampus. It further supports the view that receptor-receptor interaction, in an agonist-independent manner, can regulate key functions of glutamate synapses and associated behavior in mice. Since signaling molecules related to GPCR can be spatially structured down to the nanoscale level to ensure specificity for GPCRs[77], it further supports the view that protein-protein interaction structures the receptor nanoscale organization, downstream signaling, and essential synaptic plasticity functions. Furthermore, changes in the NMDAR membrane interactome

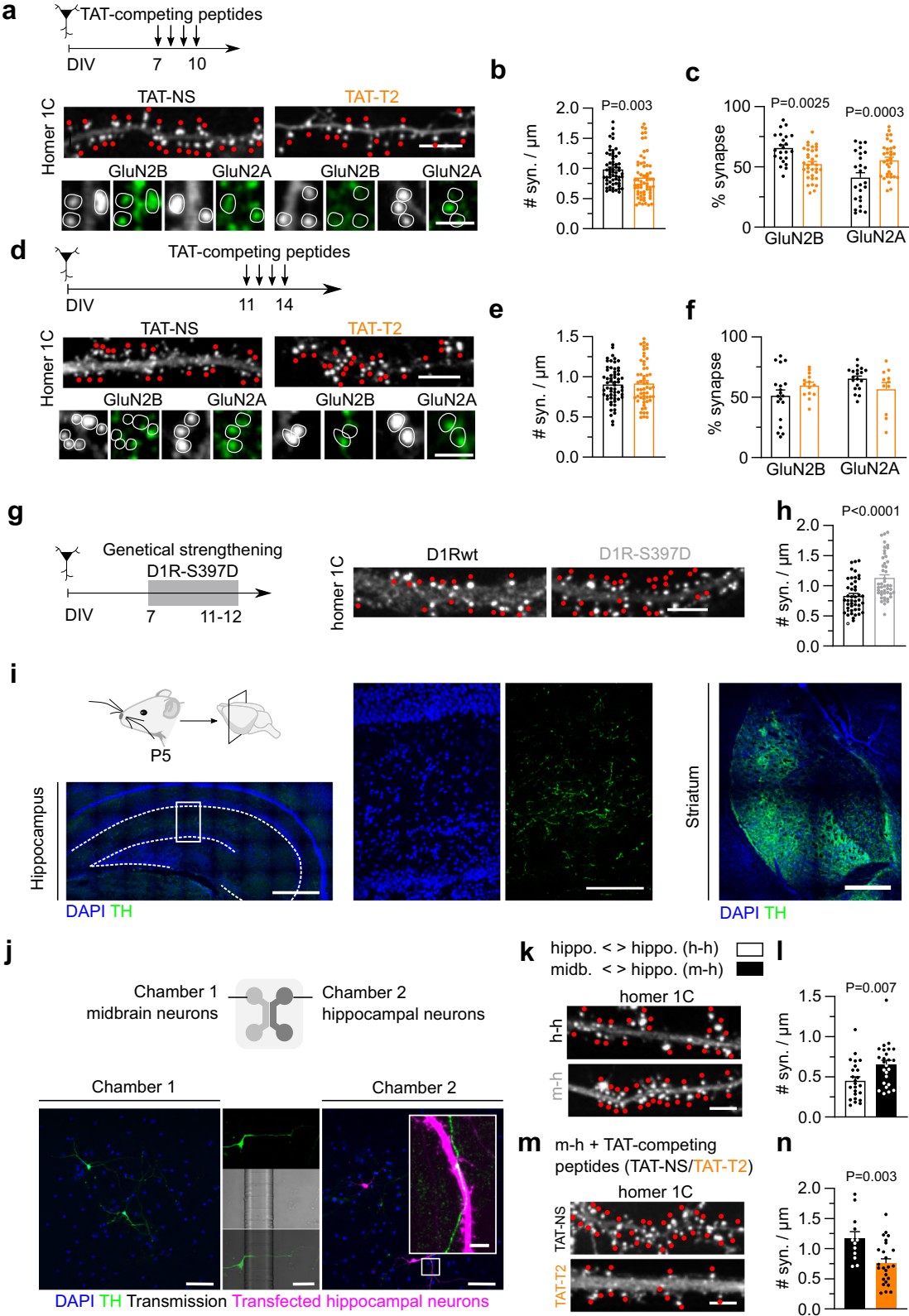

alone have been associated to the emergence of psychotic disorders[78], and more broadly receptor hetero-complexes to understand their roles in health and major brain neuropsychiatric disorders[79–84].

As a limit, our approach provides an unprecedented resolution in defining the membrane interaction between different receptors at video rate. Yet, further improving the spatial and temporal resolutions will likely shed additional lights on the protein-protein interaction.

Beyond these resolution aspects, performing these measurements with more than 2-colors will allow us to image and quantify the formation of possible heterocomplexes[41]. It also remains unknown whether NMDAR interacts in the same condition with other membrane receptors. For instance, whether mGluR5, NMDAR, and D1R form a protein complex at the surface of developing neurons is an interesting question, particularly because mGluR5 and NMDAR can directly

**Fig. 6 | GluN1-D1R interaction is necessary for synaptogenesis. a** Experimental design of the TAT-competing peptide challenge in developing immature neurons with representative images of hippocampal dendrites on which Homer 1 C cluster (synapses), GluN2A subunit, or GluN2B subunit were labeled in the presence of TAT-NS or TAT-T2 competing peptides. Scale bar, 5 and 1 μm. **b** Comparison of the number of synapses (e.g. number of Homer 1 C clusters per μm of dendrite) after treatment with TAT-NS ($n = 60$ fields) or TAT-T2 ($n = 62$) and (**c**) the percentage of synapses that are positive for GluN2B (TAT-NS, $n = 23$; TAT-T2, $n = 34$) and/or GluN2A (TAT-NS, $n = 25$, TAT-T2, $n = 33$; two-tailed Unpaired t-test). Data are presented as mean +/- SEM. Scale bar, 5 and 1 μm. **d** Experimental design alongside representative images of hippocampal dendrites on which Homer 1 C cluster (synapses), GluN2A subunit, or GluN2B subunit were labeled in the presence of TAT-NS or TAT-T2 competing peptides. Scale bar, 5 and 1 μm. **e** Comparison of the number of synapses after treatment with TAT-NS ($n = 59$ fields) or TAT-T2 ($n = 55$) and (**f**) the percentage of synapses that are positive for GluN2B (TAT-NS, $n = 19$; TAT-T2, $n = 11$) and/or GluN2A (TAT-NS, $n = 19$; TAT-T2, $n = 14$; two-tailed unpaired

t-test). Data are presented as mean +/- SEM. **g** Experimental set-up with representative images and (**h**) corresponding comparison of the number of synapses following expression of D1R-WT ($n = 44$ fields) or D1R-S397D ($n = 41$; two-tailed unpaired t-test). Data are presented as mean +/- SEM. Scale bar, 5 μm. **i** Representative images of TH immunostaining. Scale bar, (i) 500 μm and 100 μm. **j** Experimental setup with representative fluorescence images from 4 independent experiments. Scale bar, 100 μm and 5 μm. **k** Representative Homer 1 C images. **l** comparison of the synaptic density in hippocampal neurons co-cultured with hippocampal (h-h, $n = 24$ fields) or midbrain neurons (m-h, $n = 25$; two-tailed unpaired t-test). Data are presented as mean +/- SEM. Scale bar, 5 μm. **m** Representative images and (**n**) comparison of the synaptic density in hippocampal neurons co-cultured with midbrain neurons and chronically treated with competing peptides, either TAT-NS ($n = 12$ fields) or TAT-T2 ($n = 25$; two-tailed unpaired t-test). Data are presented as mean +/- SEM. Scale bar, 5 μm. Source data are provided as a Source Data file.

interact[85]. Such a large and diverse complex with ionotropic and GPCR receptors may help to confine intracellular signaling pathways to nanoscale domains[77] and integrate diverse molecular information. Technological breakthroughs that will permit to test the claim that physical interactions between various membrane receptors form a mosaic of signaling "hubs" that structure synapse formation and plasticity will thus be of prime interest.

## Methods
### Cell cultures
Cultures were kept at 37 °C – 5% $CO_2$.

**Primary neuronal cultures.** Tissue for dissociated hippocampal cultures was harvested from embryos of an unascertained mixture of sexes prevenient from gestant Sprague-Dawley rats at the age of 9–12 weeks old purchased weekly from Janvier Labs (Saint-Berthevin, France). Hippocampal cultures were prepared from embryonic stage (E18) rats. Briefly, hippocampi were dissected and collected in HBSS containing Penicillin-Streptomycin (PS) and HEPES and dissociated with Trypsin-EDTA/PS/HEPES. Cells were plated either at a density of 250 000 per 60 mm petri-dishes onto 1 mg/ml poly-L-lysine pre-coated 18 mm coverslips or at 40 000 cells per microfluidic chambers. Regarding midbrain cultures, ventral mesencephalons were dissected from E14 rats, collected in Leibovitz L-15 medium (Gibco, # 11415056), and dissociated with Trypsin-EDTA/PS/HEPES. Cells were plated at a density of 40,000 cells per chambers. Neuronal cells were maintained in Neurobasal Plus medium (Gibco, A3585911) supplemented with GlutaMAX™ (Gibco, #35050-038), B-27™ Plus (Gibco, A3653401).

**Heterologous cell culture.** COS-7 cell-line came directly from commercial sources that state for their authenticity (https://www.sigmaaldrich.com/FR/fr/product/sigma/cb_87021302). We did not perform in-house identification. All cell lines were tested negative for mycoplasma contamination. Mycoplasma testing was performed by a third-party (Eurofins) via qPCR from cell culture media. COS-7 cells were kept in Dubelcco's Modified Eagles's Medium (DMEM) supplemented with 10% fetal calf serum, 1% pyruvate and 2 mM GlutaMAX.

### Cell transfection
**Primary cultures.** Hippocampal neurons were transfected either at 7 or 10 DIV using the calcium-phosphate coprecipitation method. DNA plasmids were diluted in TE buffer (1 mM Tris-HCl pH 7.3, 1 mM EDTA) and a final concentration of 250 mM of $CaCl_2$ (2.5 M $CaCl_2$ in 10 mM HEPES, pH 7.2) were added. This mix was added dropwise to 2X HEPES-buffered saline (in mM: 12 dextrose, 50 HEPES, 10 KCl, 280 NaCl and 1.5 $Na_2HPO_4 \cdot 2H_2O$, pH 7.2). Coverslips were transferred to 12-well plate containing 250 μl/well of conditioned culture medium supplemented with 2 mM kynurenic acid. 50 μl of the precipitate solution was added

to each well and incubated for 1 h at 37 °C. Cells were then washed with non-supplemented Neurobasal medium containing 2 mM kynurenic acid and moved back to the culture dish. To prevent excitotoxicity, 50 μM of D-2-amino-5-phosphonovalerate (D-AP5) was added to the culture medium when transfecting with GluN1-NMDAR. Hippocampal neurons in co-culture with midbrain neurons through microfluidic chips were transfected using lipofectamine-2000 (Invitrogen) according to the manufacturer's recommendations. Where indicated, cells were incubated chronically with competing peptides, namely TAT-NS (YGRKKRRQRRRGSSEVILDQPVIAKPLIPALSVALSVKEEA), TAT-T2 (YGRKKRRQRRRLVYLIPHAVGSSEDLKKEEAGGIAKPLEKL) and TAT-T3 (YGRKKRRQRRRSPALSVILDYALSVVSLEKIQPVTHSGQHST) at a final concentration of 1 μM for four consecutive days or at 10 μM for 10 to 25 min.

**COS-7 cells.** Transfection with X-tremeGENE HP DNA (Roche) was done 1 day after plating. 200 μM of D-AP5 were added to the culture media when transfecting with GluN1-NMDAR. Cells were imaged 20-24 hours after transfection.

### Animals
This study was conducted in accordance with both the NIH and European Community guidelines (Directive 2010/63/EU) for the care and use of animals. Every effort was made to minimize the number of animals used and their suffering. All animals were housed and maintained on a 12-h cycle at room temperature (22 °C) and 40–70% (typically 60%) humidity with ad libitum access to food and water.

**C57BL/6 J mice.** The protocol was approved by the Experimental Animal Ethics Committee of the University of Tokyo (approval number: P29-14). A total of 18, 14 and 11 male C57BL/6 J mice at postnatal day 4, 7 and 29 with preoperative weights of 5–7 g and 20–30 g, respectively, were used in this study.

**Sprague-Dawley rats.** The protocol was approved by the Animal Care Committee of the Centre for Addiction and Mental Health (approval number: #824) of the University of Toronto as well as by the local Bordeaux Ethics Committee (APAFIS#21727-2019010918359887). A total of twice 3 male Sprague-Dawley rats at post-natal day 7 and 36 were used in this study (co-IP), and gestant Sprague-Dawley rats at the age of 9–12 weeks old were purchased from Janvier Labs, and P5 ($n = 4$), P10 ($n = 3$) and P25 ($n = 3$) animals were randomly chosen for the experimentation.

### Multi-dimensional spectral single molecule localization microscopy
Neurons were first incubated for 10 min with a mix of rabbit anti-GFP and mouse anti-Flag primary antibodies, washed and incubated for

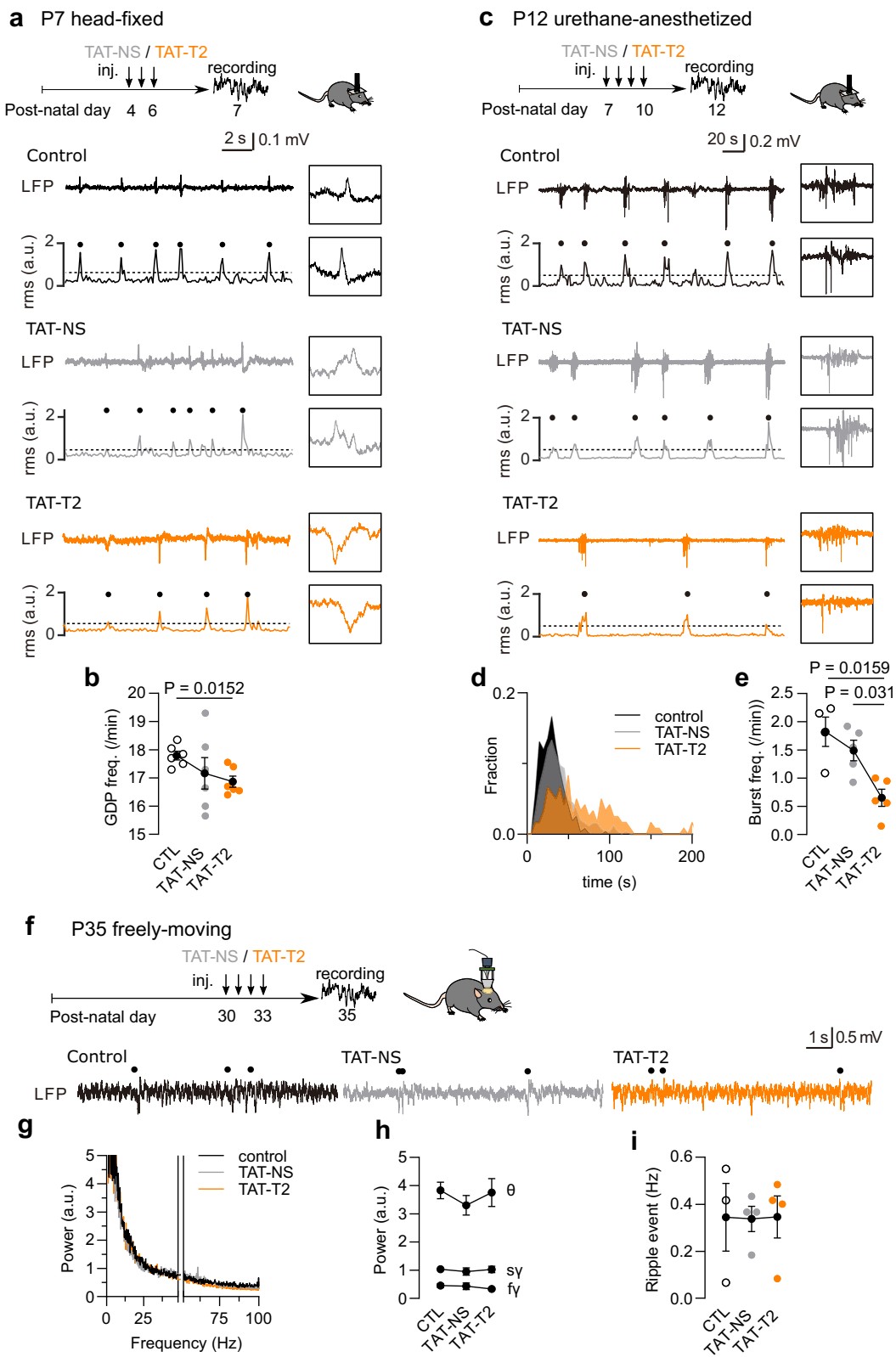

10 min with F(ab')2-Goat anti-Rabbit IgG (H + L) Secondary Antibody, Qdot 705, F(ab')2-Goat anti-Mouse IgG (H + L) Secondary Antibody, Qdot 655, and nanodiamond (Adamas Nano). All incubations were done in conditioned 1% BSA-Tyrode solution (in mM: 105 NaCl, 5 KCl, 2 MgCl2, 12 D-glucose, 25 HEPES, pH 7.4). Surface receptors diffusion were imaged for 1000 consecutive frames with an acquisition time of 50 ms (20 Hz).

**Microscopy set-up.** Our spectral microscope uses a 4Pi configuration composed of two commercially inverted microscope bodies (Nikon TiE) precisely aligned one on top of the other thanks to a (x, y, θ, φ) stage (UMS, Scientifica). The lower microscope is equipped with a high NA TIRF objective (100X Oil NA1.49, Nikon), an azimuthal TIRF/HiLo illumination device (iLAS2, Gataca Systems), a Quad band filter set (F66-04TN, AHF) and an astigmatism-based kit (manual N-STORM kit,

**Fig. 7 | GluN1-D1R interaction is required for early basal network activity in vivo. a** Experimental timeline for P7 head-fixed mice with representative LFP traces and the corresponding root mean square (RMS) in control (CTL), TAT-NS and TAT-T2 injected mice. **b** Comparison of the frequency of GDP events (*n* = 6 mice per group; two-tailed Mann–Whitney U test). Data are presented as mean +/- SEM. **c** Experimental timeline for P12 urethane-anesthetized mice with representative LFP traces and the corresponding root mean square (RMS) in control (CTL), TAT-NS, and TAT-T2-injected mice. **d** Distributions of inter-LB intervals. **e** Comparison of the frequency of LB events (*n* = 4-5 mice per group; two-tailed

Mann–Whitney U test). Data are presented as mean +/- SEM. **f** Experimental timeline for P35 freely moving mice and representative LFP traces in control, TAT-NS and TAT-T2-injected mice. Hippocampal ripple events are indicated by black dots (above the traces). **g, h** Comparison of fast Fourier Transform (FFT) plots of LFP activity at 1–100 Hz bands excluding 48-52 Hz, (*n* = 3–4 mice per group). Data are presented as mean +/- SEM. **i** Comparison of the frequency of ripple events (*n* = 3–4 mice per group; two-tailed Mann-Whitney U test). Data are presented as mean +/- SEM. Source data are provided as a Source Data file.

Nikon) which altogether enable to perform state-of-the-art 3D SMLM. The upper microscope is equipped with a high NA water dipping objective (60X, Water Dipping NA1, Nikon) and a spectral detection arm for spectral (λ) characterization of the detected single molecules. The spectral detection arm is composed of a low dispersive prism (10° edge prism, PS814-A, Thorlabs) placed in the Fourier plane of a 4 f imaging relay to convert each emitter's wavelength into a spatial displacement, laterally shifting the localization of the single emitter linearly with respect to its mean spectral emission, and a triple laser lines rejection filter (ZET 405/488/561, F67-408, AHF) to reject excitation laser light. It also integrates a ~1.5x zoom to optically match the lower (spatial) and upper (spectral) FOVs as closely as possible. Two synchronized sensitive EMCCDs (Photometrics EVOLVE 512B), one for each detection path, allow the microscope to track single emitters across 80 μm x 80 μm field-of-views. Finally, the whole 4Pi microscope is caged in a custom plexiglass heated at 37 °C (Life Imaging System) and driven by the MetaMorph software (Molecular Devices).

**Single Molecule Localization analysis.** We used PALMTracer, a custom-made software operating as a plugin of MetaMorph software, to analyze and represent the multi-dimensional (x, y, t, λ) SMLM data. It uses a combination of wavelet decomposition and 2D Gaussian fitting[86,87] to perform state-of-the-art astigmatism-based 3D single-molecule localization. Once localized, molecule trajectories are computed from the molecular coordinates of the spatial channel using a simulated annealing algorithm[88]. Files are automatically analyzed using an integrated batch engine.

**Spectrally displaced localization analysis.** The spectral determination of each localized molecule has been described here[41]. It is based (1) on the pairing of single emitter's localizations obtained on both spatial and spectral channels, and (2) on measuring the spatial shift induced by the prism inserted in the spectral detection arm. From a spectral shift calibration process, it is then possible to retrieve the mean emission wavelength of the detected single molecule. Briefly, a field transformation of the spatial localizations (lower channel) is first applied to superposed both field-of-views. Then, the localizations in the spatial and the spectral channels are paired thanks to a linear search in a pair search zone defined around the transformed spatial localization from an a-priori knowledge of the prism-induced spatial dispersion and molecules emission wavelength. The pair distance *d* is finally measured enabling to assignment an emission wavelength to the spatial localization thanks to a spectral calibration of the spectral detection arm (Supplementary Fig. 1b). The spectral calibration was performed using multicolor diffraction limited microbeads (100 nm) with well-defined fluorescence spectra, simultaneously detected on both channels. The spatial shift induced by the dispersive element inserted into the spectral detection arm is then computed measuring the distance of the localization of each emission peaks on the spectral channel from the fiducial localizations on the spatial channel after field transformation. This calibration leaded to a computed spectral shift of $-8.1 \pm 0.1$ nm/pixel (Supplementary Fig. 1c). Lastly, independent lateral drifts in either of the channels were compensated on each path separately by tracking fiducials markers of known emission

spectra to ensure robust spectrally displaced localization analysis along the entire acquisition time.

**Spectrally-informed multi-Gaussian fitting.** In order to distinguish overlapping single molecule signals, i.e. occurring when single emitters are separated by less than ~200 nm, we devised a multi-emitter fitting approach that take advantage of the localization information in the spectral channel[41] (Supplementary Fig. 1d). by increasing the robustness and accuracy of the multi-gaussian fitting process, such parameters initialization allows for monitoring receptors of different species that are simultaneously exploring the same nanoscopic environment.

**Analysis of particle-particle interactions.** A computational algorithm was developed to extract the distances separating each receptor couple in our recordings. Receptors were considered as interacting when their distance fell within the confined threshold. This threshold was set at 100 nm.

### Immunohistochemistry

Live surface staining (15 min at 37 °C) was followed by 15 min fixation in 4% paraformaldehyde (PFA) / 4% sucrose in PBS at room temperature (RT). The cells were then incubated for 15 min in PBS with 50 mM NH4Cl and blocked in PBS-1% bovine serum albumin (BSA) for 1 hour at RT. For intracellular staining, cells were fixed, permeabilized with 0.1% Triton X-100/PBS for 5 min and blocked 30 min in PBS-1% BSA. Alternatively, cells were fixed and permeabilized with ice-cold methanol. The secondary antibodies were prepared in blocking solution and incubated for 1 hour at RT. Coverslips were mounted in Fluoromount media and kept at 4 °C until imaging.

When needed, cells were, prior to fixation, co-incubated for 25 min at 37 °C with primary antibodies and various inhibitors or 5 min with 50 μM of glutamate or KCl (at 50 mM or 2.5 mM).

### Microfluidic devices production

Microfluidic molds were fabricated on glass by soft lithography with the UV-curable adhesive NOA81 (Thorlabs) which resulted in a positive relief pattern of the microfluidic chip. A mixture of PDMS (Sylgard 184) with curing agent (10 to 1 ratio) was poured onto the positive replicate, degassed under vacuum before reticulated over-night at 70 °C. The resulting negative replica-polymer was detached, punched to create four reservoirs, cleaned and dried in sterile conditions. Finally, the polymer print was bonded onto a 1 mg/ml poly-L-lysine coated glass coverslip and stored at 37 °C until cell seeding.

### Epifluorescence confocal image acquisition and analysis

**Spinning disk.** Images were acquired using a high-speed spinning disk confocal unit equipped with an electron multiplying charge-coupled device (EMCCD) camera (Photometrics QuantEM 512SC) through either a 20x objective (Leica, HC PLAN APO, 0.7 NA) and/or a 63x oil objective (Leica, HCX HPL APO CS, 1.4-0.6 NA). Hardware was controlled with MetaMorph software (Molecular Devices). All images were analyzed with ImageJ 1.53c (National Institute of Health, USA). For the Sholl Analysis, hippocampal neurons were transfected with a GFP-encoding plasmid at DIV 7, and hippocampal dendritic trees were

reconstructed using the ImageJ' plugin "SNT". The overlap between D1R and GluN1-NMDAR was defined as the fraction of GluN1 cluster area that overlap with D1R cluster.

**Calcium imaging.** Neuronal cells were transfected with GCaMP6f together with either D1R-wt, D1R-dT2 or D1R-S397D at DIV 7 and imaged 2-3 days post-transfection. For isolation of NMDAR-dependent $Ca^{2+}$ transients, neurons were incubated in $Mg^{2+}$-free Tyrode's solution containing $5\,\mu M$ nifedipine and $5\,\mu M$ bicuculline for 15 min before imaging. Two time-lapse images of 3000 frames were acquired at 20 Hz, one before and one 5 min after incubation with D-AP-5 at $50\,\mu M$. Calcium activity was analyzed as previously described[89]. Briefly, mean normalized fluorescence i.e. $\Delta F/F$ was calculated by subtracting each value with the mean of the previous 5-s values lower than $P_{50}$ ($\mu$) and dividing the result by $\mu$. $\Delta F/F$ traces were smoothened by convoluting the raw signal with a 10-s squared kernel and positive calcium transients were automatically defined based on a threshold set at 5*SD of the AP-5 trace.

### dSTORM

Images were acquired using a Nikon Ti eclipse system equipped with a Perfect Focus System (PFS), an azimuthal TIRF arm (Gataca Systems, Massy, France), and an Apo TIRF 100x NA1.49 oil-immersion objective and an Evolve EMCCD camera (Photometrics, Tucson, USA) with a final pixel size of 65 nm. This system is equipped with a Ti-S-ER motorized stage controlled by MetaMorph software (Molecular Device, USA). Samples were illuminated in TIRF mode and images were obtained with an exposure time of 20 ms with up to 40,000 consecutive frames. Imaging was carried out at room temperature in a closed Ludin chamber (Life Imaging Services, Switzerland) using a pH-adjusted extracellular solution containing oxygen scavengers and reducing agents. Multicolor fluorescent microspheres (Tetraspeck, Life Technologies) were used for lateral drift correction. Super-resolution images were reconstructed with PALMTracer and protein-clustering into nano-size clusters i.e. nanodomains was obtained using the SR-Tesseler method[90]. Segmentations of the clusters were performed by applying a threshold of twice the average density $\delta$ of the whole dataset, with a minimum area of 7 and a minimum number of localizations of 5. Clusters' nanodomains were identified by applying a threshold of one time the average density of each cluster (0.4 minimum area, 25 minimum number of localizations). To analyze GluN1 enrichment at post-synapses (considered as homer 1 C puncta), the average density of detections was divided by the average density of extra-synaptic detections.

### Frequency-domain-based FRET-FLIM microscopy

COS-7 cells were co-transfected with carboxyl terminally tagged GluN1-GFP together with HA-GluN2A and with the carboxyl terminally tagged D1R (WT or S397D)-mCherry in a proportion of 1:1:1, unless stated otherwise. mCherry alone was used as a FRET-negative control. Cells were imaged with an HCX PL Apo 63x oil NA 1.4 objective using an appropriate GFP filter set. Cells were excited using a sinusoidally modulated 3 W 478 nm LED (light-emitting diode) at 36 MHz under wild-field illumination. Emission was collected using an intensified CCD LI2CAM camera (Lambert Instrument BV, Groningen, The Netherlands). Lifetimes were calibrated using a solution of erythrosin B that was set at 0.086 ns. The lifetime of the sample is determined from the fluorescence phase-shift between the sample and the reference from a set of 12 phase settings using the manufacturer's LI-FLIM software.

### In vivo electrophyisological recording

**Peptide administration.** TAT-NS and TAT-T2 peptides (3 mg/kg, i.p.) were daily administered for 3–4 consecutive days. For P4 mice, peptide administration was performed for 3 days and an electrophysiological recording was performed under a head-fixed condition at P7 (termed

P7 head-fixed). For P7 mice, peptide administration was performed for 4 days and an electrophysiological recording was performed under a urethane-anesthetized. For P29 mice, an electrode assembly was first implanted into the hippocampus and peptide administration was performed for 4 days from P30 and recordings were obtained at P12 (termed P35 freely moving). Control mice were not injected with any drugs.

**Surgery and electrophysiological recording.** For electrophysiological recording from P7 head-fixed mice, the mice were anesthetized with isoflurane gas (0.5–2.5%) and restrained with their head held in place by a metal plate. A craniotomy was performed to create a rectangular hole ($3.0 \times 1.0\,mm^2$) centered at 1.5 mm posterior and 1.5 mm lateral to the bregma using a metal cutter, and the dura was surgically removed. Two 32-gauge needles were implanted in the bone above the cerebellum to serve as ground and reference electrodes. A silicon probe that consisted of 64 recording sites (Buzsaki 64, NeuroNexus) was inserted into the brain at a speed of $5\,\mu m/s$ so that the final depth of the electrode tip in the brain was $1300\,\mu m$. The electrodes were allowed to stabilize at their final position for 10 min before recording began. To aid in the reconstruction of the tracks left by the probe, the backside of the probe was coated with a DiI solution (80 mg/ml, Invitrogen).

For electrophysiological recording from P12 urethaneanesthetized mice, the mice were anesthetized with urethane (1.5 g/kg, i.p.) and the same procedures applied to the P7 head-fixed mice were performed with a craniotomy centered at 1.8 mm posterior and 1.7 mm lateral to the bregma and the final depth of the electrode tip in the brain ranged from 1500 to $1800\,\mu m$.

For electrophysiological recording from P35 freely moving mice, the mice were implanted with an electrode assembly at P29. For electrode implantation, the mice were anesthetized with isoflurane gas (1–2%). A craniotomy with a diameter of ~2 mm was performed using a high-speed drill, and the dura was surgically removed. Two stainless-steel screws were implanted in the bone above the cerebellum to serve as ground and reference electrodes. An electrode assembly that consisted of 4 tetrodes, which was created using a 3D printer (Form 2, Formlabs), was stereotaxically implanted above the right hippocampus (1.8 mm posterior and 1.5 mm lateral to bregma). The tip of the electrode bundle was lowered to the cortical surface, and the electrodes were inserted 1.8–2.0 mm into the brain at the end of surgery. The electrodes were constructed from 17-$\mu m$-wide polyimide-coated platinum-iridium (90/10%) wire (California Fine Wire), and the electrode tips were plated with platinum to lower electrode impedances to 150–300 $k\Omega$ at 1 kHz. Electrophysiological data were sampled at 2 kHz and filtered between 0.1 and 500 Hz for at least 15 min using a Cereplex direct recording system (Blackrock).

**Histological analysis to confirm electrode locations.** At the end of the recording in P12 mice, the silicon probe stained with DiI was carefully removed from the brain. All mice were perfused intracardially with cold 4% PFA in 25 mM PBS and decapitated. After dissection, the brains were fixed overnight in 4% PFA. For P12 mice, the brains were rinsed in PBS and coronally sectioned at a thickness of $100\,\mu m$ by a vibratome. For P35 mice, the brains were equilibrated with 30% sucrose in PBS, coronally sectioned at a thickness of $50\,\mu m$ by a microtome, and counterstained with cresyl violet. The positions of electrodes were confirmed by identifying the corresponding electrode tracks in histological tissue. When electrode positions were not clearly visible, electrodes showing apparent LB events at P12 or ripple events at P35 were considered as electrodes located inside the hippocampus.

**LFP data analysis.** For LFP recording data from P7 head-fixed mice, the 20-min LFP traces were band-pass filtered at 1–50 Hz and the root

mean square (RMS) was computed with a bin size of 100 ms and giant depolarizing potentials (GDPs) were detected when the RMS exceeded 3 standard deviations above the mean. The minimum intervals between neighboring GDPs were set to be 1 s.

For LFP recording data from P12 urethane-anesthetized mice, to reduce 50-Hz humming noise, a 40-60 Hz notch filter was applied to the LFP data. The RMS of the 60-min LFP traces was computed with a bin size of 1 s and large-amplitude burst (LB) events were detected if the RMS exceeded a threshold of 1 mV/s.

For LFP recording data from P35 freely moving mice, the power spectra of LFP traces during 60-s quiescent periods at a moving speed of less than 2 cm/s were calculated by fast Fourier transformation at frequencies ranging from 1 to 100 Hz. The power of LFPs in the following sub-frequency bands were calculated: theta (6–10 Hz), slow gamma (20–40 Hz), and fast gamma (60–100 Hz). For the detection of ripples, LFP signals were band-pass filtered at 150–250 Hz and the RMS was calculated in the ripple-band with a bin size of 20 ms. Ripple events were detected when the RMS exceeded 3 standard deviations above the mean.

### Tissue preparation
P5 animals were anesthetized with pentobarbital (300 mg/kg) and transcardially perfused with 4% PFA. Whole brains were removed and fixed overnight in 4% PFA, cryoprotected by immersion in 30% sucrose solution and sliced into 20 μm thick coronal sections on a microtome-cryostat (Leica CM3050S).

### Biochemistry
**Western Blot.** Brain samples were snap-frozen in liquid nitrogen and stored at −80 °C. Neuronal cultures and brain samples were homogenized with TET buffer (20 mM Tris pH 8 − 1 mM EDTA −1.3% Triton X-100) complemented with proteases inhibitors), incubated on ice 10 min and centrifuge 10 min at 10,000 $g$ to remove debris. The protein concentration of all samples was simultaneously determined using Pierce BCA Protein Assay Kit (Thermo Fisher Scientific). Either 4 or 10 μg of protein was loaded in 4-20 % precast SDS-polyacrylamide gel electrophoresis and transferred to a nitrocellulose membrane (Bio-Rad, Hercules, USA). The membrane was blocked in 5% non-fat milk Tris-buffer saline (TBS)/0.1% Tween-20 (TBST) at R.T. for 1 hour. Primary antibodies were diluted in 0.5 % milk TBST and incubated O.N. at 4 °C under agitation. Incubation with corresponding secondary antibody was performed for 1 h at R.T. Specific protein stain was revealed with SuperSignal West Femto Maximum Sensitivity Substrate detection kit (Pierce, Thermo Fisher Scientific Inc., Cambridge, UK) and total membranes were scanned using a Li-COR Odyssey-Sc imaging system. Quantification of band intensity was performed using Odyssey software and it was normalized to tubuline staining. Full scan blots are available in the Source Data file.

**Co-immunoprecipitation.** The co-immunoprecipitation assay was performed as previously described[50]. Briefly, rat brain tissues were homogenized in lysis buffer (50 mM Tris-HCl, 150 mM NaCl, 2 mM EDTA, 0.5% sodium deoxycholate, 1% NP-40, 1% Triton X-100, 0.1% SDS, Protease inhibitor cocktail (Sigma-Aldrich, 1:100, pH 7.4) on ice. Then the samples were gently shaken at 4 °C for 1 hour and centrifuged at 10,000 $g$ for 10 min. The supernatant was collected as the protein extract. The concentration of samples was measured using Pierce BCA Protein Assay. For co-immunoprecipitation experiments, 500 – 700 μg protein extract was incubated with protein A/G plus agarose (25 μl per sample; Santa Cruz Biotechnology, catalog number: sc-2003) at 4 °C for 1 hour, and then the supernatant was incubated together with new protein A/G plus agarose in the presence of primary antibodies against D1R (2 μg) or D2R (2 μg) or control IgG (1–2 μg) at 4 °C for 12 hours with gentle shaking. Pellets were washed, boiled for 5 min in SDS sample buffer and 2-Mercaptoethanol and subjected to SDS-PAGE. A total of

50-100 μg of protein extract was used as a control in each experiment. Full scan blots are available in the Source Data file.

### Antibodies

| Reference | | Provider | Dilution |
|---|---|---|---|
| **Primary antibodies** | | | |
| rabbit polyclonal anti-GFP | #A-6455 | ThermoFisher Sc. | 1/500 or 1/10 000 |
| mouse monoclonal anti-Flag | #F1804 | Sigma-Aldrich | 1/500 or 1/10 000 |
| mouse monoclonal anti-TH | #MAB318 | Merck Millipore | 1/1000 |
| mouse anti-CK1 alpha | #sc-75582 | Santa Cruz | 1/1000 |
| mouse monoclonal anti-beta-tubulin | TUB2.1 | Sigma-Aldrich | 1/5000 |
| Rabbit anti-GluN2A | Clone A12W | Merck Millipore | 1/1000 (WB) |
| rabbit anti-GluN2A | custom-made | Agrobio | 1/200 (IF) |
| rabbit anti-GluN2B | custom-made | Agrobio | 1/200 (IF), 1/1000 (WB) |
| rabbit anti-D1R | 17934-1-AP | Proteintech | 2 μg |
| rabbit anti-D2R | 55084-1-AP | Proteintech | 2 μg |
| rabbit anti-NR2A | NB300-105 | Novus Biologicals | 1/1000 |
| rabbit anti-NR2B | ab65783 | Abcam | 1/1000 |
| Chicken anti-MAP2 | ab5392 | Abcam | 1/5000 |
| Mouse anti-synapsin 1 | #106011 | Synaptic System | 1/1000 |
| Mouse anti-PSD95 | MA1-046 | ThermoFisher Sc. | 1/500 (IF), 1/1000 (WB) |
| Rabbit anti-DBH | ab209487 | Abcam | 1/2000 |
| **Secondary antibodies:** | | | |
| goat anti-mouse alexa fluor 488 | #A11001 | ThermoFisher Sc. | 1/1000 |
| Donkey anti-mouse alexa fluor 647 | #A31571 | ThermoFisher Sc. | 1/1000 |
| goat anti-rabbit alexa fluor 647 | #A21244 | ThermoFisher Sc. | 1/1000 |
| Goat anti-chicken 488 | #A11039 | ThermoFisher Sc. | 1/1000 |
| anti-mouse H + L HRP | #715-035-150 | Jackson Immunoresearch | 1/5000 |
| goat-anti-rabbit IgG (H + L) highly cross-absorbed secondary antibody Alexa Fluor Plus 800 | #A32735 | ThermoFisher Sc. | 1/5000 |
| Alexa Fluor 790 AffiniPure Goat-anti-Rat IgG (light chain specific) | #112-655-175 | Jackson Immu-noResearch | 1/1000 |
| Alexa Fluor 790 AffiniPure Goat-anti-Rabbit IgG (light chain specific) | #115-655-174 | Jackson Immu-noResearch | 1/1000 |
| **Quantum Dots:** | | | |
| F (ab')2-Goat anti-Rabbit IgG-coupled Qdot655 | #Q11422MP | ThermoFisher Sc. | 1/50000 |
| F (ab')2-Goat anti-Rabbit IgG-coupledQdot705 | #Q11461MP | ThermoFisher Sc. | 1/50000 |

### Statistical analysis
No statistical methods were used to predetermine sample size. Sample size was based on previous publications with similar models and experiments. To ensure replicability, results are derived from at least

three independent experiments. No data were excluded from the analysis. All statistical tests were performed using GraphPad Prism. Datasets were analyzed for normality and parametric or non-parametric statistical test (two-tailed) were used accordingly. Test details and statistical outcomes are reported in the relevant figure and figure legends.

### Reporting summary
Further information on research design is available in the Nature Portfolio Reporting Summary linked to this article.

## Data availability
Data and resources are available on request from the corresponding author. Source data are provided as a Source Data file. Source data are provided with this paper.

## Code availability
Codes used for this paper are available on request from the corresponding author.

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

## Acknowledgements

We thank Julien Dupuis, François Maingret, and Olivier Nicole for insightful discussions, Hélène Gréa for training on microfluidic device, the Bordeaux Imaging Center for support in microscopy and in particular Christel Poujol and Magalie Mondin for advice and discussions, the IINS in vivo facility for animal housing. We would also like to thank the Cell Biology Facility, especially Delphine Bouchet, Constance Manso, Elodie Cougouilles, Morgane Meras, Léa Villetelle, Emeline Verdier, Christelle Breillat, and Rémi Sterling, for molecular and cellular tool productions and general cell biology activity management. This work was supported by The National Center for Scientific Research (CNRS), Agence Nationale de la Recherche (Projects DynHippo and DopamineHub to L.G.; Nano-PlanSyn and soLIVE to J.B.S), Human Frontier Science Program (RGP0019/2016 to L.G., Y.I., F.L.), Fondation pour la Recherche Médicale, European Research Council Synergy grant (ENSEMBLE, #951294 to L.G.). We also acknowledge France-BioImaging infrastructure supported by the French National Research Agency (ANR-10-INBS-04 to L.G. and J-B.S.).

## Author contributions

N.B., C.B., G.E.S., N.K., Y.N., T.S., T.Y., P.S., and M.P-P. conducted the experiments and collected the data. C.B., R.G., J-B.S. developed the MS-SMLM. P.S. and F.L. performed co-immunoprecipitation experiments. V.S. supervised microfluidic experiments. N.B., C.B., G.E.S., N.K., Y.N., T.Y., P.S., and M.P-P. analyzed the data. N.B., G.E.S., T.S., Y.I., and L.G. designed the study and experiments. N.B. and L.G. wrote the paper, which all authors helped to revised.

## Competing interests

The authors declare no competing interests.
