## [Peer Review File · Nature Communications]

Non-canonical interplay between glutamatergic NMDA and dopamine receptors shapes synaptogenesisReviewers' comments and responses

Reviewer #1

The manuscript by Dr Benac and colleagues from the laboratory of Dr Groc uses a variety of techniques and approaches to examine a previously characterized interaction between the NMDAR and Dopamine receptor type 1. The authors use a clever modification of the single particle tracking methods they have previously used to great effect to demonstrate that these proteins interact. They then use a variety of other approaches to probe how the interaction between these proteins change during development, is modified by neuronal activity, and how it may alter neuronal connectivity. Overall, this is an interesting and important study that breaks new ground. However, there are a few issues that need to be addressed.

We thank the reviewer for the positive comments and suggestions that helped strengthening the study (see below).

One important issue is that although the authors highlight the MS-SMLM approach they develop, it seems to be used only in the first two data figures. The data provided by MS-SMLM is important, and it would be of significant interest if the authors were able to show how manipulations that alter the apparent interaction between the NMDAR and the D1R (activity blockade (TTX) or activation of the glutamate receptors) alters the kinetics of the interaction between these proteins. These experiments would be informative for the mechanisms involved.

Response: We agree with the reviewer that deciphering how agonist binding or neuronal activity alters the interaction between the two receptors is of great interest. It is already known that bath application of glutamate or D1/5R agonist greatly alter the membrane dynamics of NMDAR and D1R. These bulk changes would greatly confound our analysis of the interaction, changing the interaction probability, location of interaction, eventually generating false-colocalization of immobilized receptors. Thus, to provide a convincing and physiologically-relevant answer to this question one needs to perform both MS-SMLM together with local, and not global, modulation (e.g. uncaging of glutamate or dopamine). Indeed, this setting will be important to control in a precise temporal and local window the biophysical property of the interaction under the microscope. Yet, this experiment is very challenging and we were not able to perform it in the required highest standard and fully controlled manner. Several difficulties need to be solved. For instance, interactions occur in a stochastic manner so uncaging agonist just before a putative interaction is a gambling game. Also, combining MS-SMLM imaging with local uncaging in live neurons is a technological challenge at the microscopic level. We thus feel that providing strong and solid data to answer the question is beyond the scope of the study and our current capacity to develop the required experiment setup.

The experiments in figure 4 are a bit confusing. The authors show the effects of TTX and glutamate on the overlap of NMDAR and D1R puncta. Then attempt to determine the receptor subtype mediating the effects of glutamate. They show a small effect of inhibiting mGluRs and no significant effects of blocking with APV and NBQX. Surprisingly they do not show whether blocking both AMPAR and NMDAR results in a change. This experiment needs to be done.

Response: We fully agree and have performed these experiments that are now included in the revised manuscript. Our results show that dual blockade of AMPAR with NBQX and NMDAR with AP-V do not change D1R-GluN1-NMDAR colocalization (new Fig. S4).

Then based on a single mGluR antagonist (LY) the authors conclude a specific effect. Additional experiments are needed. They need to show agonists of the mGluRs mimic the effects of glutamate, and that another mGluR blocker blocks the effects of glut. To conclude these effects are due to a specific mGluR. The authors need to show that antagonists of other mGluRs have no effect on the interaction.

Response: We agree with the reviewer and have performed additional series of experiments. In the revised version of the manuscript, we further confirmed the role of mGluRs in mediating the effect of ambient glutamate on the interaction between D1R-GluN1-NMDAR since we show that the initial effect of LY341495 was reproduced with MTEP, which is another antagonist of mGluRs. In addition, the effect of glutamate was mimicked by DHPG, which is an agonist of mGluRs. The specificity of the mGluR effect was further confirmed by showing that depolarizing neurons with KCl (50 mM), which strongly increase fast synaptic transmission and thus glutamate

receptor activation did not change D1R-GluN1-NMDAR colocalization. These data have been added in the revised manuscript (Figure 4, Fig. S4).

Similar experiments are needed to implicate CK1.

Response: We perform an additional series of experiments to confirm the role of CKI in regulating the interaction by reproducing the effect of CKI-7 with another CKI antagonist, IC-261 (Fig. S4). We further tested the specificity of the CK by blocking CKII with TMCB. This did not alter the interaction between D1R-GluN1-NMDAR, further supporting a specific role of CKI (Figure 4). However, to our knowledge, there is no specific agonist of CKI that will further clarify its specific role. Our new data have been included in the revised manuscript.

The authors conclude that “the GluN1-NMDAR-D1R interaction is upregulated... through recruitment of mGluR5 and CK1”. The authors need to show evidence of this mechanism. Practical experiments could include showing that there is an inducible co-IPs of mGluRs and CK1 with the NMDAR-D1R complex or an inducible association of these proteins by ICC or MS-SMLM. Even if the authors were to limit their claims, additional experiments are needed, and more work to explore whether the proposed complex forms.

Response: The possibility that NMDAR, D1R, mGluR5, and CK1 form a complex has never been investigated. Our data neither support, nor discard this possibility as the functional crosstalk between these proteins does not require *per se* a direct interaction. As suggested by the reviewer, a co-IP experiment will indeed provide evidence, or not, that these proteins assemble in a complex. This experiment and appropriate control are however difficult to perform, and we believe beyond the scope of the study. Already, the co-IP between D1R and GluN1 has recently turned to be difficult to perform due to the lack of appropriate antibodies. Moreover, with an ICC experiment it will be very difficult to demonstrate the presence of a complex since CK1 is a cytosolic protein present along the dendritic tree. For all these technical challenges, the proposed experiments are not feasible in the timeframe of this study. More importantly, we have now clarify the message of the paper, stating that our data neither support, nor discard the possible presence of such protein complex. We mention at the end of the discussion that it will be interesting in the future to explore possible protein complexes, such as NMDAR/D1R/mGluR5/CK1/others.

In figure 5, the authors appear to assume that the only synapses that form on a neuron are on spines. Particularly during development, this is not the case. A DIV10, most synapses are formed on the shaft of the dendrite. By DIV15 there is a large increase in spine density. The authors need to determine the location of pre-synaptic sites in these young neurons to claim that calcium transients and clusters are extra synaptic. This could be done with post-hoc staining for release sites (bassoon) or labeling for presynaptic markers.

Response: We agree with the reviewer that, early on, most spines are located on the shaft (this is indeed illustrated on both Fig. 3A and 3F @ 9-DIV). However, by 12-DIV that was the age of the neurons used for the calcium imaging experiments, the vast majority of postsynaptic densities were located onto protrusions (Fig. S6 of the revised manuscript). This has been clarified in the revised manuscript to avoid confusion.

In figure 6, the authors claim that a microfluidic culture system they use demonstrates a dopamine-dependent effect on the spine density (synapse density would require staining for both pre-and postsynaptic markers). They claim these effects are due to changes in the interaction between the NMDAR and D1R. However, they provide no evidence that the culture conditions change the NMDAR-D1R interactions, or that there are different levels of dopamine in these cultures (other than experiments showing effects of SKF-38393 on the interaction).

This claim should be directly tested, showing that dopamine treatment reduces the interaction and spine density, that blocking the D1R increases the interaction and spine density, and that agonists of other dopamine receptors has no effect on the interaction or spine density.

Response: Our data show that the presence of dopaminergic axons in the hippocampal network increases the number of glutamatergic synapses. We performed an additional series of experiments and analysis to determine the effect of TH-positive axons on hippocampal neuron dendritic tree. We show that the dendritic tree of hippocampal neurons exposed to TH-positive axons were more developed (Fig. S8), consistent with the well-documented trophic effect of dopamine on neuronal development.

Yet, preventing the receptor interaction with a competing peptide altered the number of synapses irrespective to the absence or presence of TH-positive axons. Thus, these data support the view that the synaptogenic role of the receptor interaction runs in a “dopamine-independent” manner. We have now edited the revised manuscript with the additional set of data and to clarify our message.

Minor:

The authors state that neuronal activity is required for synapse formation; however, numerous studies have shown that this is not the case. For instance, synapses form without synaptic transmission (*munc-13* knockouts). The authors must provide evidence/references for this statement or remove this claim.

Response: We have removed and edited this claim from the revised manuscript.

The authors do not indicate how the TAT peptides were developed, and what controls were done to insure their selectivity and specificity. Are these the peptides published by this group in Ladepeche et al. 2013? If the peptides are new, more information is needed about their generation, testing, and characterization.

Response: This is now specified in both the Results and Material and Methods sections. The peptides were the same as previously published.

The title of figure 3 – States that the ‘interaction is up-regulated’. The authors do not show upregulation but that the interaction is higher in younger neurons. To show upregulation, they would need to show that it increases from a lower level to a higher one via a regulatory mechanism. Are the authors saying that before DIV10, the interaction is lower? As is these data show that between DIV10 and DIV15 the interaction decreases.

Response: We have edited this claim and clarified the message. We indicated that the values are increased compare to mature neurons.

The authors should better define vague definitions like immature or mature – particularly since they determine these stages using an elegant approach. The authors should state the age range used (DIV 10 and DIV 15). It seems like it is more interesting that the change in the interaction occurs over a short period of only five days.

Response: We have addressed this comment by precisizing the age of the neurons of the two groups. We however kept the denomination *immature* and *mature* as we believe it helps in simplifying the reading of the manuscript.

Reviewer #2

I was asked to comment on the section describing the effects on early development.

I strongly suggest to remove all these results from the manuscript, since they do not at all show what the authors claim (“Preventing the interaction in the neonate, and not adult, hippocampus strongly alters in vivo neuronal network activity”).

1. In this in vivo experiment, the authors attempted to disrupt D1R267 GluN1-NMDAR interaction by injecting TAT-T2 peptide into the peritoneal space of mice before recording the hippocampal local field potential (LFP) activity at two different ages: at the beginning and the end of synaptogenesis (P7 and around P30) and examined the effect on network activity. During the first week of life, newborn rats show relatively long silence of neocortical activity interrupted by transient “early spindles”. These spindle bursts represent the first and only organized network pattern in the thalamocortical system, triggered by spontaneous muscle twitches, motor patterns analogous to human fetal movements (“baby kicks”). However, by day 7, these “trace alternans” patterns are replaced by more mature and continuous activity in the drug-free rat (Khazipov et al., *Nature* 2004). What the authors show in Fig. 7 is a typical LFP pattern in deep urethane anesthesia, similar to patterns in the adult brain under urethane. Urethane affects the release of both GABA and glutamate (Mochol et al., *PNAS* 2015; Torao-Angosto et al., *Front Syst Neurosci.* 2021), thus urethane anesthesia (or any anesthesia) is a poor choice to address the issue.

Response: In revision, we made a series of additional experiments to test the effects of competing (TAT-T2) peptides in “non-anesthetized” mice on P7 within the synaptogenesis window (Fig. 7a-b). We confirmed that they spontaneously generated giant depolarizing potentials (GDPs) in the hippocampus, which represent synchronous burst firing of neuronal populations in the developing brain. Notably, administration of competing TAT-T2 peptide, but not TAT-NS peptide, significantly reduced the GDP frequency (Fig. 7a-b). These results provide direct

in vivo evidence that, even under more natural conditions without anesthesia, D1R-GluN1-NMDAR interaction is required for proper neuronal network activity at early stages.

To facilitate data comparisons, we decided to leave the previous datasets from P12 urethane-anesthetized mice (Fig. 7c-e), showing that competing TAT-T2 peptides reduced large-amplitude burst (LB) electrical events, which also likely represent synchronous burst firing of neuronal populations in the developing brain.

In addition, as shown in our previous manuscript, we demonstrated that chronic administration of competing peptides in adolescent mice (between P30-33) had no effect on either LFP oscillatory patterns or sharp wave/fast oscillatory ripples (Fig. 7f-g-h-i).

Finally, we also performed a series of experiment using another anesthetic molecule, pentobarbital. However, this drug strongly inhibited neuronal network activity, preventing a useful and thoughtful analysis. This additional dataset was however not added in the revised manuscript.

2. *The authors compared P7 animals under urethane anesthesia with drug-free adults. This is like comparing apples and bananas.*

Response: As answered above, we have now added data using P7 awake animals (without any anesthesia).

3. *Fig. 7. The power comparisons are made at theta, slow and fast gamma bands. However, anesthesia changes these frequencies. Under urethane anesthesia theta is between 1.5 to 4 Hz, present typically after tail pinching (arousal), whereas in the behaving rodent it is typically 6-10 Hz.*

Response: As the reviewer suggested, anesthesia has been shown to change the range of theta oscillations. We thus did not compute the LFP power from P12 urethane-anesthetized mice and simply presented the event frequency of burst activity (Fig. 7c-e).

4. *Line 275: "Chronic administration of competing TAT-T2 peptide significantly reduced LB frequency and increased the inter-event intervals" What is the difference? Frequency is 1/time interval, right?*

Response: Yes. The unit on the Y-axis is "/min".

5. *Line 279: "These results demonstrate in vivo that D1R-GluN1-NMDAR interaction is indeed required for proper neuronal network activity at early stages, with no detectable effect on the spontaneous hippocampal network activity at adult stage." No, the results certainly do not support this claim.*

Response: The text has been edited according the new dataset and following the reviewer comment.

Discussion

Line 294: "Preventing the interaction in the neonate, and not adult, hippocampus strongly alters in vivo neuronal network activity." This single sentence is all the comments the authors present in the Discussion on early development. A bold statement without any discussion of potential mechanisms.

Response: This statement has been edited to avoid confusion.

Reviewer #3

Comments for the Author:

The authors analyze here the interaction between NMDA receptors (NMDAR) and dopamine D1 receptors (D1R), using a number of quantitative assays, from multidimensional spectral single molecule-localization microscopy to in vivo analyses of neuronal network activity. They conclude that this interaction is important for the developing brain. The work is very well designed and executed, but it is unclear whether the advances it brings are sufficient for publication in Nature Neuroscience.

We thank the reviewer for the positive evaluation of our careful work.

As the authors explain in the introduction, the interaction of D1R with NMDARs has been studied for more than two decades. It has been observed in native tissue (hippocampi) by immunoprecipitation already in 2002 (Lee et al., Cell; reference 18 of the current manuscript), and its functional aspects have been discussed and analyzed in many works so far. The interacting components from the receptors have been well described, by many laboratories. The team of Dr. Groc has already published the co-clustering of D1Rs and NMDARs in 2013 (Ladepêche et al., PNAS; reference 21 of the current manuscript). Changes in the receptor behaviors were observed after manipulating their interaction with TAT peptides, similar to the present manuscript, with the authors concluding that “the D1R–NMDAR interaction bidirectionally regulates the surface distribution and dynamics of D1R and NMDAR” (title of Fig. 2 in the respective work).

The main points of the work can be summarized as follows:

- Imaging of both NMDARs and D1Rs simultaneously, using quantum dot tracking. The results confirm previous findings relating to this interaction (Figs. 1 and 2 of the manuscript), and add the conclusion that this interaction is stronger in immature neurons (Fig. 3).
- The interaction is related to activity, confirming previous observations from the field (Fig. 4).
- The interaction is related to the clustering of the molecules (Fig. 5), thus confirming the work from Ladepêche et al./reference 21.
- Perturbing the interaction by use of TAT peptides (previously characterized in the literature) affects synaptogenesis (Fig. 6) and brain activity in very young animals (Fig. 7), but not in older animals.

The last point is novel, with this manuscript suggesting, for the first time, that the D1R–NMDAR interaction is mainly important for synapse formation, but this may be an effect of the experimental procedure used. Synapses are presumably far less stable in 10 days-old animals than in 30-day old animals. Disturbing synaptic receptors for 3 days is probably more likely to change synaptic organization in the young animals than in the older ones. It is possible that the use of TAT peptides for a longer period (as 10 days) would also lead to perturbations in the older animals.

Response: We agree and thank the reviewer for this proposed experiment that was performed and included in the revised manuscript. Treating mature hippocampal neurons daily for 8 consecutive days with 1 μ M of competing peptide had no effect on the number of glutamatergic synapses (see Fig. S7). This result further strengthens the role of D1R–GluN1–NMDAR interaction at early stage in favoring synaptic maturation.

Overall, while I find the work to be technically excellent, it is difficult to accept its conclusion that it offers a first visualization of the D1R–NMDAR receptor interaction, which has been determined in native tissues, by biochemical means, already 21 years ago, and which has been well described by many groups, including the team of the last author. I therefore find the work more suitable for a more specialized journal.

Response: We agree that the interaction between GluN1–NMDAR–D1R has been described more than two decades ago, but respectfully disagree with the reviewer point-of-view for the following reasons. It is fair to state that the presence as well as the role of the weak receptor-receptor interactions (NMDAR–DopamineR, as well as many other heterocomplexes) has been consistently questioned in the field (difficulty to reproduce the biochemical experiments demonstrating the presence of membrane heterocomplexes), casting doubt on their existence in live neuronal membrane. In addition, the methods commonly used to describe that interaction, e.g. co-immunoprecipitation or proximity ligation assay, are unable to answer fundamental questions regarding the interaction itself. For instance, the duration, the occurrence of the interaction, the location, as well as the regulation of the interaction were simply unknown. Our SMLM method allows to specifically monitor surface receptors diffusion and interaction at the nanoscale level and with millisecond temporal resolution, unveiling previously unknown key parameters of the interaction. We have now edited the revised manuscript in order to better explain our view on the question, although once again we perfectly understand the point-of-view of the reviewer.

I would suggest one change to the presentation of the experiments: Figures 4, 5 and 6 rely on extremely small zoom-in panels for many of their points. It would be useful to present the entire images, or at least larger frames, in a supplementary file. For example, the panels selected for control and CKI-7 in Fig. e are extremely different, albeit the quantification only shows an approximately 40% difference between the two conditions. This type of question would be better served by including larger images.

Response: We agree and edited the figures accordingly.

In the present work, Bénac, Saraceno, Butler and colleagues investigate the dynamics of dopamine 1 (D1R) and NMDA (NMDAR) receptor heteromerization and its role in shaping synaptogenesis in hippocampal neurons. The main novelty of this work relies on the development of an elegant imaging approach, the multidimensional spectral single molecule-localization microscopy (MS-SMLM), which allowed for the first time to study, both spatially and temporally, the dynamics of lateral diffusion and interactions of the D1R and the GluN1 subunit of NMDARs in immature and mature cultured hippocampal neurons. They show that the transient D1R-NMDAR interactions are up regulated in immature neurons relative to mature ones and try to identify the signaling events modulating these interactions. Finally, they develop approaches to evaluate the roles of D1R-NMDAR heteromerization in modulating synaptogenesis in cultured neurons and hippocampal network activity in vivo to conclude that D1R-NMDAR interactions play a functional role in the developing brain. Although this study is the first to dynamically monitor D1R-NMDAR interactions over time, some additional experiments are needed to strengthen the robustness of the main findings, clarify the molecular mechanisms involved in the modulation of this interaction, and conclude on their role in vivo in mice.

We thank the reviewer for their positive comment and suggestions that strengthen our study.

- Figure 1: Describes the MS-SMLM set up and dynamics of D1R-NMDAR in cultured hippocampal neurons. Although this extremely informative and powerful approach represent a significant breakthrough in the field, the parameters to identify episodes of interactions (less than 200 nm proximity between receptor partners) are not explain and the resolution of this approach compared to other is not discussed.

Response: We apologize for the lack of information in the main text (the information were provided in the Supplementary Material). We have now addressed this comment in the revised manuscript by increasing the explanation of the imaging approaches and the parameters used to define interaction.

- Figure 2: the authors studied the interaction of a genetically-modified D1R lacking the t2 domain, which binds to the GluN1, and showed that the occurrence and duration of D1R-dT2 interaction are decreased. However, D1R-dt2 is still able to interact with GluN1 (according to the parameters used to define when an interaction occurs). This may be related to the fact that D1R-dT2 is still able to bind GluN2A NMDAR subunits. The same experiments should be performed with a version of D1R lacking both the T2 domain and the T3 fragment, which is responsible for D1R-GluN2A interaction (Lee et al. Cell 2002). In these conditions, D1R-NMDAR interactions should be further reduced, or not detectable anymore. This would strengthen our understanding of the mechanisms controlling these interactions and eventually shed light on the accuracy and resolution of MS-SMLM.

Response: We thank the reviewer for the comment. It was indeed difficult to conclude that the interaction we described was solely controlled by the T2 sequence (GluN1-D1R). To clarify this important point, we performed a series of experiments in which we investigate the role played by both the T2 and T3 domain on the interaction between GluN1-NMDAR-D1R. We used FRET imaging in COS-7 cells in order to get rid of all the neuronal confounding parameters. Cells were either transfected with D1R together with GluN1 and GluN2A or GluN2B subunits. We observed similar FRET signals between GluN1-GluN2A-D1R and GluN1-GluN2B-D1R (Fig. S2), suggesting that the presence or absence of GluN2A subunit did not alter NMDAR-D1R interaction. Furthermore, applying TAT-T3 peptide, which prevent the D1R-GluN2A-NMDAR interaction, did not alter the FRET signal between GluN1-2A::D1R whereas the TAT-T2 peptide did so (Fig. S2). These data indicate thus that, in our setting, the T3 domain seems to play a negligible role on the NMDAR-D1R interaction when compared to the T2 domain. Noteworthy, young and immature hippocampal neurons that were used in our experiment express a low level of GluN2A subunit (Fig. S2), emphasizing further that the effect we report here are mainly, if not exclusively, due to the interaction between D1R and GluN1 subunit. We have included these new dataset and further clarify the revised manuscript.

The result section related to figure 2 mention that “Spatially, D1R-GluN1-NMDAR interaction were highly labile”, which is supported by the data, but it is also said that these interactions “occurred randomly onto the dendritic shaft (no evidence of dedicated interaction hot spots), with very rare observation of such events in spines”, however, the experiments allowing such claims are not presented in this figure.

Response: We have added a representative reconstruction of GluN1-NMDAR and D1R dual-color tracking experiment where we highlighted the localization of the events (see supplementary Fig. 1d). In addition, we clarify that this note was based on qualitative observation, and not specific analysis.

- Figure 3: In panel g, it is not clear why D1R co-immunoprecipitation with GluN1 has not been performed. To make the link with figure 6, D1R-GluN1, D1R-GluN2A and eventually D1R-GluN2B (even though it is not a direct interaction) should be performed. Furthermore, from the illustrative immunoblots, there is a higher D1R-GluN2A co-IP at immature stages but the efficiency of D1R is also higher. No information is available regarding the quantifications used but the amount of GluN2A co-immunoprecipitated with D1R should be normalized to the efficacy of D1R IP.

Response: The co-IP between GluN1 and D1R is currently problematic as the former antibody used to bind D1R is no longer available. We have tested many alternatives and antibodies without a satisfying outcome so far. For this technical reason, we performed the co-IP with another GluN subunit.

- Figure 4: Describes pharmacological approaches to assess the signaling pathways regulating D1R-NMDAR interactions. To make the link between mGluR5 stimulation, CKI activity and D1R phosphorylation on S397, it is necessary determine whether glutamate induces D1R-GluN1 heteromerization through an increase of CKI activity and D1R phosphorylation. This implies to investigate CKI activity and D1R pS397 upon glutamate in the presence or not of the mGluR5 antagonist. This will precise the signaling pathways regulating D1R-NMDAR interaction and bring novelty with regard to the already known role of D1R pS397 in modulating D1R-GluN1 interaction (Woods et al. Journal of Molecular Neuroscience 2005; Scott et al. PNAS 2006).

Response: We agree that directly assessing CK1 activity and D1R-pS397 levels will bring key insight into the signaling cascades involved. Yet, to our knowledge, such tools are not available. Nonetheless, we confirmed the role of mGluRs in mediating the effect of ambient glutamate on the interaction in a CK1-dependent manner. Indeed, the effect of LY341495 were reproduced with MTEP, which is another antagonist of mGluRs, and the effect of glutamate were mimicked by DHPG, which is an agonist of mGluRs (Fig. 4, Fig. S4). Such specific role of mGluRs were further confirmed by depolarizing neurons with 50 mM of KCl, which leads to the activation of synaptic receptors (e.g. AMPAR and NMDAR) as it did not change D1R-GluN1-NMDAR colocalization (Fig. 4). Finally, the inhibition of CK1 abrogated the effect of DHPG on the interaction, further highlighting a functional interplay between glutamate, mGluR, and CK1 in regulating the interaction between GluN1-NMDAR and D1R (Fig. 4).

- Figure 5: Study the impact of D1R-NMDAR interaction on GluN1 expression at synaptic/extrasynaptic sites. Panel a shows illustrative images of D1R-GluN1 overlap in the presence of D1R-wt, D1RS397D and D1R-dT2. Instead of showing the same images as in Fig.4, these images should also include the staining of the synaptic marker to accompany the panel d.

Response: We have now added representative images of the Homer 1C staining alongside the superresolved image of surface GluN1-NMDAR (figure 5d).

What is not clear to me is the interpretation of the fact that D1R-S397D, which promotes D1R-NMDAR, favors GluN1 expression at extrasynaptic sites, while the deletion of the t2 domain (inhibiting D1R-NMDAR) has no effect.

Response: We agree and faced the same questioning. Indeed, one could expect a bi-directional effect with, on one hand, an increased retention of extrasynaptic NMDARs in D1R-S397D, and on the other hand, a decreased retention of extrasynaptic NMDARs in t2 domain. Yet, we only observed the increased retention with D1R-S397D. Although we don't have experimental data to explain these observations, one could speculate that there are some compensatory mechanisms that stabilize and clusterize extrasynaptic NMDARs in the t2 domain condition. Further in-depth investigation, beyond the scope of this study, will be necessary.

- Figure 6: In panel i, the use of TH as a marker of dopamine afferents in the hippocampus is not accurate as this will label also serotonergic and noradrenergic fibers. An alternative would be to stain dopamine beta hydroxylase. This also apply to experiments carried out in microfluidic devices (panel j).

Response: We, of course, agree with the reviewer that, although commonly used to stain for dopaminergic neuron, TH staining alone does not distinguish dopaminergic from other monoaminergic fibers. We therefore

made this experiment in the microfluidic condition since we draw conclusion from this experiment (Figure 6j-k-l). We see negative DBH staining which was not surprising since we cultured ventral mesencephalon of E14-embryos that do not contain noradrenergic neurons (DBH-positive) from the locus coeruleus or serotonergic neurons from the raphe. This has been clarified in the revised manuscript and an additional material is provided (Supplementary Fig. 8). Regarding the hippocampal slice image, our intent was purely qualitative as the dopaminergic and noradrenergic innervations have been thoroughly described over the past decades.

In panel 6i, a TAT-T2 peptide is used to disrupt D1R-GluN1 interaction and TAT-NS is used as a control peptide (this also applies to figure 7). The specificity of the interventional approach is questionable since the same peptide has been used to alter D1R-D2R heterodimers (see Perrault et al. European Neuropsychopharmacology 2016 as an example [10.1096/fj.14-254037](https://doi.org/10.1096/fj.14-254037)). The use of alternative approaches to disrupt D1R-NMDAR for comparison with the TAT-T2 effects would strengthen the results obtained both in cultured neurons and in vivo (figure 7).

Response: The question of the D1R-D2R complexes T2 sequence indeed overlaps partly with one of the sequence identified in the formation of D1R/D2R heterocomplexes. It should first be noted that the existence of the D1/D2 interaction has been highly debated and remain controversial (Frederick et al., Mol. Psychiatry, 2015). In addition, multiple and distant domains of the D1R C-tail appear to be involved in the interaction with D2R. However, the mechanism underpinning this GPCR-GPCR complex is not understood and it is quite unclear whether of the domain (not targeted by the t2 peptide) is enough for the interaction. Second, and most importantly, D2R are very poorly express in the hippocampus so the presence of such putative complexes is more than unlikely.

- Figure 7: Panel a, b, it is necessary to show that the regimen of TAT-T2 administration used is indeed able to alter D1R-NMDAR in vivo in the hippocampus.

Response: Systemic administration, including intra-peritoneal injection, of such TAT-peptides is commonly used to break in vivo protein interaction into the brain. We previously demonstrated the central effect of such competing TAT-peptides injected intraperitoneal at neuronal and behavioral levels (Su et al., Neuron, 2014). To note, similar TAT-based peptides injected intraperitoneal were also validated in different brain conditions (e.g. Stanic et al., Nature Communications, 2015). This has now been clarified in the revised manuscript.

It seems that experiments carried out involve 3 groups of mice: control (i.e no TAT-peptide), TAT-NS and TAT-T2. LFP traces should be shown for all groups and the number of animals per group (n = 4-5) should be increased because the statistical power of analysis is not very high. This is particularly important for panel 7d given that the TAT-NS control peptide tends to lower burst frequency with regard to the control group. Of note, the interpretations of these in vivo is too hippocampus-centered since TAT-peptide were systematically administered, therefore potentially altering D1R-NMDAR in the whole brain.

Response: We have now strengthened this section of manuscript by performing 2 new series of experiments: i) head-fixed recording at P7 pups, and ii) anesthetized (pentobarbital) P12 pups. In order to compare the awake young and adult mice dataset, we now included this set of data in the main figure (Figure 7). The data strengthen our former claim and support an early hippocampal activity disruption by the competing peptide. Regarding the change in anesthesia, pentobarbital was however silencing the hippocampal network, precluding any useful analysis.

As a general comment, the manuscript clearly lacks technical information and technical justification on multiple occasions. As an example, multiple experiments rely on overlap of fluorescent signals. There is not clear information about what is considered as an overlap and how it is measured.

Response: We have now addressed this comment in the revised manuscript by increasing the amount of information regarding technics in both the Results and Material and Methods sections.

REVIEWER COMMENTS

Reviewer #1 (Remarks to the Author):

The authors have effectively responded to the majority of my comments. The manuscript will be of interest to the neuroscience community. I think some effort could be made to place the role of the effects that authors observe with other synaptogenic factors which also act at a similar point in development (eg LRRRTMs, Ephs, NLGNs etc). Regardless, the data present are of high quality and impact.

Reviewer #2 (Remarks to the Author):

In my previous Review, I suggested to remove the electrophysiological experiments because they were substandard in quality. I still think that removing Fig. 7 in fact would improve the overall quality of the paper.

The authors added experiments in 6 head-fixed mice and measured thresholded LFP changes in control conditions (control and TAT-NS) and after 3 days of TAT2 treatment. While they found a significant difference between saline control and TAT2, there was no difference between TAT-NS and TAT2. Thus, the proper conclusion is there was no drug-induced effect.

Even if the difference was significant, one group (p7) was tested in head-fixed condition whereas the other (p35) under unconstrained situation. The incidence and magnitude of the large population bursts are subject to brain-state changes, with a predominance in NREM sleep. However, how the stressful head-fixing changes brain state is not known.

The authors also include results under urethane anesthesia but because that drug alters so many features of brain dynamics there is no way to compare it to drug-free adults. How many sessions and mice were used?

LFP bands in the adult are super sensitive to brain state. No information is provided whether the recordings were induced during immobility, exploration or sleep and how the drugs affected these states. From the raw LFP it appears that the mice were not moving. But what is the relevance of measuring theta band power, if no theta is expected to be present? No information is provided about the number of animals involved or the statistics performed. In short, the physiological experiments do not provide convincing results.

Reviewer #3 (Remarks to the Author):

The authors have replied convincingly to my comments, and I suggest that the manuscript be published.

Reviewer #4 (Remarks to the Author):

The referee would like to thank the authors for the considerable amount of work they have performed to address the vast majority of concerns raised. This revised manuscript has been substantially improved to the point that I believe it is now acceptable for publication.

Two minor points:

- Figure 1: Panel 1d is not mentioned in the text. Fig.3b: overlay pre-post synaptic staining.
- Figure 3: Panel 3a should include and overlay MAP-2/PSD-95/Synapsin staining to appreciate the localization of PSD-95 relative to synapsin that has been used to quantify the number of synapses.

Reviewers' comments and responses

Reviewer #1

The authors have effectively responded to the majority of my comments. The manuscript will be of interest to the neuroscience community. I think some effort could be made to place the role of the effects that authors observe with other synaptogenic factors which also act at a similar point in development (eg LRRTMs, Ephs, NLGNs etc). Regardless, the data present are of high quality and impact.

We greatly thank the reviewer for the positive comments. We have added a sentence on the role of other synaptogenic molecules (discussion, last paragraph).

Reviewer #2

In my previous Review, I suggested to remove the electrophysiological experiments because they were substandard in quality. I still think that removing Fig. 7 in fact would improve the overall quality of the paper. The authors added experiments in 6 head-fixed mice and measured thresholded LFP changes in control conditions (control and TAT-NS) and after 3 days of TAT2 treatment. While they found a significant difference between saline control and TAT2, there was no difference between TAT-NS and TAT2. Thus, the proper conclusion is there was no drug-induced effect. Even if the difference was significant, one group (p7) was tested in head-fixed condition whereas the other (p35) under unconstrained situation. The incidence and magnitude of the large population bursts are subject to brain-state changes, with a predominance in NREM sleep. However, how the stressful head-fixing changes brain state is not known.

Response:

The mice at this age (P7) are naturally almost immobile and have not yet developed apparent state-dependent LFP patterns, such as theta oscillations and ripples in the hippocampus. As a marker of organized neuronal activity at this age, we then focused on the well-described GDPs. Due to the obvious difficulty of applying electrophysiological recordings under freely moving conditions to mice of this age, we conducted recordings under head-fixed conditions. We cannot precisely quantify the level of stress imposed by the head-fixed condition, which was obviously similar between all animal groups. These information have now been added in the revised manuscript (Methods section).

The authors also include results under urethane anesthesia but because that drug alters so many features of brain dynamics there is no way to compare it to drug-free adults. How many sessions and mice were used?

Response:

In the series of experiments using urethane, the number of mice used were 4-5 per group. For recordings, there was no session in the recording of P12 animals. We conducted recordings for at least 15 min from each individual mouse. All these information have now been added in the revised manuscript in order to clarify our procedure.

LFP bands in the adult are super sensitive to brain state. No information is provided whether the recordings were induced during immobility, exploration or sleep and how the drugs affected these states. From the raw LFP it appears that the mice were not moving. But what is the relevance of measuring theta band power, if no theta is expected to be present? No information is provided about the number of animals involved or the statistics performed. In short, the physiological experiments do not provide convincing results.

Response:

During recordings, mouse behavior and health were monitored by a video-camera placed above the recording chamber. For each animal, sequences including movement periods (more than ~2 cm/s) quantitatively defined from the video recordings and important movement-related electrical noise in LFPs were excluded for analysis. We selected only sequences with stable recordings from immobile states for our analysis.

For the theta power, we agree that the meaning of theta during immobility state make little, if any, sense. It was never our intent and we did not comment the meaning of these dataset in that sense. Yet, we would like to stress out that there was no change in theta power in all our conditions confirms that behavioral states of the mice were

stable over the recording periods and not variable across groups, suggesting that mice were likely in similar immobile states.

Reviewer #3

The authors have replied convincingly to my comments, and I suggest that the manuscript be published.

We greatly thank the reviewer for the positive evaluation and past constructive comments.

Reviewer #4

The referee would like to thank the authors for the considerable amount of work they have performed to address the vast majority of concerns raised. This revised manuscript has been substantially improved to the point that I believe it is now acceptable for publication.

We thank the reviewer for their positive comment and suggestions that strengthened our study.

Two minor points:

- Figure 1: Panel 1d is not mentioned in the text. Fig.3b: overlay pre-post synaptic staining.

The error/typo has been corrected.

- Figure 3: Panel 3a should include and overlay MAP-2/PSD-95/Synapsin staining to appreciate the localization of PSD-95 relative to synapsin that has been used to quantify the number of synapses.

We did not perform 3-color immunostaining in this experiment, precluding the colocalization of MAP-2/PSD-95/Synapsin.

REVIEWERS' COMMENTS

Reviewers' comments and responses

We thank once again the reviewers for their positive and enthusiastic comments. The manuscript has been edited according the checklist (provided in the submission).